META-RESEARCH

# Questionable research practices may have little effect on replicability

**Abstract** This article examines why many studies fail to replicate statistically significant published results. We address this issue within a general statistical framework that also allows us to include various questionable research practices (QRPs) that are thought to reduce replicability. The analyses indicate that the base rate of true effects is the major factor that determines the replication rate of scientific results. Specifically, for purely statistical reasons, replicability is low in research domains where true effects are rare (e.g., search for effective drugs in pharmacology). This point is under-appreciated in current scientific and media discussions of replicability, which often attribute poor replicability mainly to QRPs.

**ROLF ULRICH\* AND JEFF MILLER\***

**\*For correspondence:** ulrich@uni-tuebingen.de (RU); miller@psy.otago.ac.nz (JM)

Most sciences search for lawful data patterns or regularities to serve as the building blocks of theories (e.g., *Bunge, 1967*; *Carnap, 1995*; *Popper, 2002*). Generally, such data patterns must not be singular findings (i.e., chance findings) but instead be replicable by other researchers under similar conditions in order to be scientifically meaningful (*Popper, 2002*, p. 23). With this fundamental scientific premise as background, it is understandable that many researchers have become concerned that a surprisingly large number of published results cannot be replicated in independent studies and hence appear to represent chance findings or so-called false positive results (*Baker and Penny, 2016*; *Ioannidis, 2005b*; *Pashler and Harris, 2012*; *Simmons et al., 2011*; *Zwaan et al., 2018*). For example, only less than 30% of results in social psychology and about 50% in cognitive psychology appear to be reproducible (*Open Science Collaboration, 2015*). Similarly, the replication rate of 21 systematically selected experimental studies in the social sciences published between 2010 and 2015 in *Nature* and *Science* was estimated to be only about 62% (*Camerer et al., 2018*). Low replication rates have also been reported in medical

research (*Begley and Ellis, 2012*; *Ioannidis, 2005a*; *Prinz et al., 2011*): for example, researchers at the biotechnology firm Amgen tried to confirm findings in 53 landmark studies in preclinical cancer research, but were able to do so for only six cases (*Begley and Ellis, 2012*). The Reproducibility Project: Cancer Biology was set up to further explore the reproducibility of preclinical cancer research (*Errington et al., 2014*).

## Possible causes of low replication rates

Understanding the causes of these shockingly low replication rates has received much attention (e.g., *Button and Munafò, 2017*; *Pashler and Harris, 2012*; *Schmidt and Oh, 2016*), and various possibilities have been discussed. First, scientists may fabricate data to support their hypotheses. However, surveys indicate that this is probably not a major cause because the prevalence of scientific fraud is low—probably smaller than 2% (see *Fanelli, 2009*; *Gross, 2016*; *Stroebe et al., 2012*).

Second, *Benjamin et al., 2018* recently argued that the traditional $\alpha$ level of 5% is too

large and thus produces too many false positives. These authors suggested changing the critical $\alpha$ level to 0.5%, because this "would immediately improve the reproducibility of scientific research in many fields" (p. 6). Although this change would decrease the false positive rate, it would also *increase* the proportion of false negatives unless there were substantial increases in sample size (*Fiedler et al., 2012*).

Third, another important factor seems to be the typically low statistical power in psychological research (*Button and Munafò, 2017*; *Stanley et al., 2018*). Some have reported average power estimates as high as 50% to detect a correlation of 0.2 (corresponding to Cohen's $d = 0.43$) in the field of social-personality psychology (*Fraley and Vazire, 2014*). In a large survey of over 12,000 effect sizes, however, *Stanley et al., 2018* reported that median power was about 36% and that only 8% of all studies had a power of about 80%. Even lower median power of about 21% has been reported for studies in the neurosciences (*Button et al., 2013*). Low power within a research area reduces replicability for purely statistical reasons, because it reduces the ratio of true positives to false positives.

Fourth, the percentage or "base rate" $\pi$ of true effects within a research area strongly influences the replication rate (*Miller, 2009*; *Miller and Ulrich, 2016*; *Wilson and Wixted, 2018*). When $\pi$ is small, the relative proportion of false positives within a given research domain will be high (*Ioannidis, 2005b*; *Oberauer and Lewandowsky, 2019*), and thus the replication rate will be low. This is easily seen: for $\pi = 0$ the relative proportion of false positives is 100%. In contrast, for $\pi = 1$, no false positives can occur so this proportion is zero. Consequently, replication rates must be higher when the base rate is relatively high than when it is low. For example, *Wilson and Wixted, 2018* have argued that the fields of cognitive and social psychology differ in the base rate of real effects that are investigated, which they call the "prior odds." On the basis of the results obtained by the *Open Science Collaboration, 2015*, they estimated base rates of $\pi = 0.20$ for cognitive psychology and $\pi = 0.09$ for social psychology, and these estimates are consistent with the finding that the replication rate is lower for social than cognitive psychology. Alternative analyses of replication rates and prediction markets also suggest similarly low base rates of about 10% (*Dreber et al., 2015*; *Johnson et al., 2017*; *Miller and Ulrich, 2016*). More generally, it is reasonable to assume that base rates differ between discovery-oriented research and theory-testing research (*Lewandowsky and Oberauer, 2020*; *Oberauer and Lewandowsky, 2019*).

Finally, a certain percentage of false positive results is an unavoidable by-product of null hypothesis testing, and, more generally, of any uncertain dichotomous-choice situation in which one is required to choose between two alternatives, such as "accept" or "reject" a vaccine as beneficial in the fight against a certain infectious disease. In such situations, many have argued that replication rates are low because questionable research practices (QRPs) used by scientists chasing after statistically significant results produce an excess of false positive results beyond the usual nominal significance level of 5% (*Ioannidis and Trikalinos, 2007*; *John et al., 2012*; *Simmons et al., 2011*). Such practices violate not only the basic assumptions of the null hypothesis significance testing (NHST) framework but also those underlying decision making within the Bayesian framework, where researchers could analogously use QRPs to obtain large Bayes factors (*Simonsohn, 2014*).

Hence, a bias toward publication of significant results or large Bayes factors provides a strong incentive to use QRPs (*Bakker et al., 2012*), especially when competing for academic promotion (*Asendorpf et al., 2013*) or grant funding (*Lilienfeld, 2017*). A survey conducted by *John et al., 2012* identified several such practices, and the most frequent ones can be grouped into four categories (a) A researcher may capitalize on chance by performing multiple studies and using *selective reporting* of a significant result. For example, the researcher may conduct several similar experiments until one finally yields the hoped-for significant result, and then the researcher only reports the results of the one study that 'worked', putting negative results into the file drawer (*Rosenthal, 1979*). There is convincing evidence that researchers conduct several studies to examine a hypothesis but only report those studies that yielded confirming results (*Francis, 2014*; *Francis et al., 2014*). (b) A researcher may measure multiple dependent measures and report only those that yield significant results. For example, a

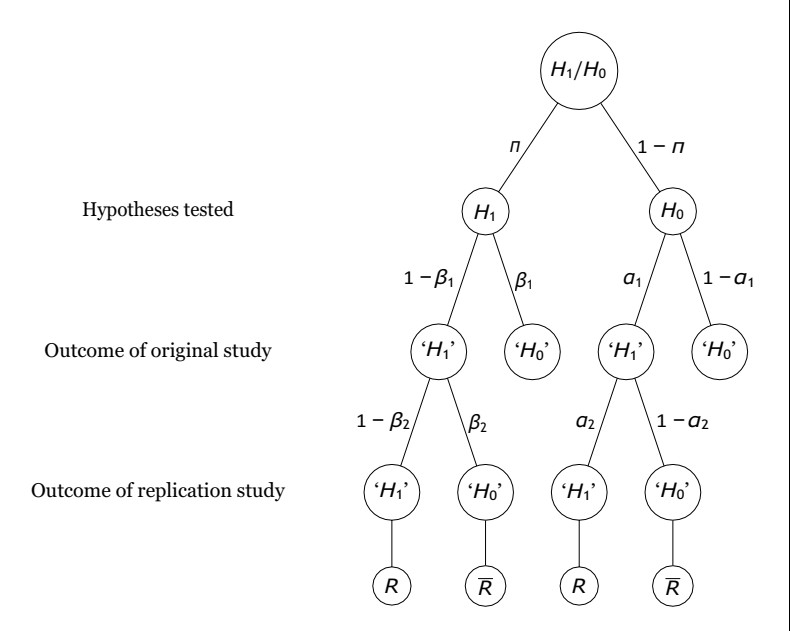

**Figure 1.** Probability tree of the replication scenario. The base rates of examining an alternative hypothesis $H_1$ or a null hypothesis $H_0$ are $\pi$ and $1 - \pi$, respectively. The statistical power and the Type 1 error rate of the original study are $1 - \beta_1$ and $\alpha_1$. There are four possible outcomes of an original study, with the researcher deciding to reject the null hypothesis (i.e., '$H_1$') in two outcomes and failing to reject it (i.e., '$H_0$') in the other two. If $H_1$ is true, the outcomes associated with these decisions are called true positives and false negatives. By contrast, if $H_0$ holds, they are called false positives and true negatives. Replication studies replicate original studies that reported a significant positive result. The statistical power and the Type 1 error probability of the replication study are $1 - \beta_2$ and $\alpha_2$, respectively. The replication study may either reject $H_0$ (which denotes a successful replication of the original positive result, $R$) or fail to reject it (which denotes a failure to replicate the original result, $\overline{R}$).

neuroscientist could record brain activity in hundreds of distinct brain areas and report the results only for those that were sensitive to a specific experimental manipulation (*Vul et al., 2009*). With 10 moderately correlated dependent measures (i.e., $\rho = 0.2$) and one-tailed tests, for example, this strategy of *multiple testing* raises the rate of false positives from 5% to 34%. (c) A researcher may monitor data collection, repeatedly testing for significant results, and stop data collection when a significant result is attained. This strategy of *data peeking* can easily raise the rate of false positives up to 20% (*Simmons et al., 2011*). (d) Finally, *selective outlier removal* can also turn a nonsignificant result into a significant one (*Ulrich and Miller, 1994*). For example, if an initial analysis produces nonsignificant results, a researcher may try different criteria for excluding outliers in the hope of getting significant results after the data have been 'cleaned'.

With all four of these QRPs as well as other ones, the researcher exploits the degrees of freedom present in the research process to achieve a statistically significant result—a practice that has been referred as "*p*-hacking" (*Simonsohn et al., 2014a*). This clearly inflates the rate of false positives, which would intuitively be expected to decrease replicability. What has received considerably less attention, however, is that *p*-hacking also increases the statistical power for detecting true effects, as noted recently by *Witt, 2019*—a side-effect of *p*-hacking that might be termed *power inflation*. Since increasing power also increases replication rates, the influence of QRPs on power tends to counteract its influence on Type 1 error rate with respect to overall replicability. A quantitative model is therefore needed to assess the size of *p*-hacking's overall effect on replicability.

In this paper, we consider in detail the prevailing claim that QRPs are a major cause of low replicability. However, *Francis, 2012a* has noted the converse problem that in some circumstances QRPs can artificially increase replication rates. Specifically, this can happen when researchers use QRPs to significantly replicate their previous findings—usually with conceptual replications—to strengthen their theoretical position. Reanalyses of results from multi-experiment papers suggest that this does happen, because the rate of successful replication is unrealistically large in view of the studies' power (e. g., *Francis et al., 2014*; *Francis, 2012b*). For example, when the power of a single experiment is 0.36, the probability that a series of five experiments would all result in positive outcomes is $0.36^5 = 0.006$, so such a series of published findings would be too good to be true (i. e., an excess of positive results). Such a pattern would suggest the operation of one or more QRPs; for example, negative results may have been unreported, that is, put in the researcher's file drawer. This situation could be called "motivated replication" and it is different from the situation in which an unbiased researcher tries to replicate a significant result, as in the Open Science Replication Project (*Open Science Collaboration, 2015*), We shall focus on the situation with unbiased replications and assess the extent to which QRPs can reduce the rate of these.

In the present study, we develop a quantitative model of replication rate that simultaneously takes into account $\alpha$, power, the base rate of true effects, and *p*-hacking. This model allows us to assess the relative contributions of these factors to the replication rate, with a focus on the

influence of QRPs. In contrast, the combined effects of *p*-hacking on Type 1 error rate and power have not previously been modelled at all, and previous studies have generally considered the effects of these factors on replicability one at a time (e.g., $\alpha$, power, base rate), making it difficult to see their relative contributions. Knowledge of the relative contributions of these different factors would increase our understanding of why the observed replication rate is so low and thus be useful in guiding efforts to improve the situation. Because the various different *p*-hacking strategies reviewed above may have different impacts on the replication rate, we conducted separate analyses for each strategy.

## Statistical analysis of the replication scenario

The analyses in this manuscript address replication scenarios in which researchers conduct direct replications of studies that reported a statistically significant positive outcome. An example is the Open Science Replication Project (*Open Science Collaboration, 2015*), in which many independent research teams conducted high-powered studies attempting to directly replicate published results. *Figure 1* depicts these scenarios together with all statistically relevant parameters that must be taken into account when computing the rate of replicating significant results (*Miller, 2009*; *Miller and Ulrich, 2016*; *Miller and Schwarz, 2011*). First, each original study tests either a true effect (i.e., $H_1$ is true) or a null effect (i.e., $H_0$ is true), with base rate probabilities $\pi$ and $1 - \pi$, respectively, and these probabilities—sometimes called "pre-study probabilities" (*Ioannidis, 2005b*) or "prior odds" (*Wilson and Wixted, 2018*)—may vary across research fields (*Wilson and Wixted, 2018*). If the original study tests a true effect, its statistical power is $1 - \beta_1$ and the Type 2 error probability is equal to $\beta_1$. Thus, the compound probability of examining a true effect and rejecting the null hypothesis is $\pi \cdot (1 - \beta_1)$; this outcome is called a "true positive." In contrast, if the original study tests a null effect, its Type 1 error probability is $\alpha_1$. Thus, the probability of testing a null effect and falsely rejecting $H_0$ is $(1 - \pi) \cdot \alpha_1$; this outcome reflects a "false positive." Note that, in keeping with accepted procedures for null hypothesis testing, we categorize studies as rejecting the null hypothesis or not based on an all-or-none comparison of computed *p*-values relative to an $\alpha$ level cutoff.

Such a discrete categorization is, for example, how most journals currently evaluate statistical results in publication decisions and how replication success or failure has mainly been operationalized in empirical studies of replication rates (*Camerer et al., 2018*; *Open Science Collaboration, 2015*).

Only true positives and false positives enter into replication projects. The statistical power $1 - \beta_2$ and Type 1 error probability $\alpha_2$ of the replication studies might differ from those of the original study, especially because replication studies are usually designed to have much higher power than the original studies. Thus, the compound probability of examining a true effect that yields a significant effect in the original and in the replication study is $\pi \cdot (1 - \beta_1) \cdot (1 - \beta_2)$, whereas the compound probability of examining a null effect and finding significance in both the initial study and the replication study is $(1 - \pi) \cdot \alpha_1 \cdot \alpha_2$. From the above compound probabilities, the rate of replication of initially significant results, *RR*, can be computed as

$$RR = \frac{\pi \cdot (1 - \beta_1) \cdot (1 - \beta_2) + (1 - \pi) \cdot \alpha_1 \cdot \alpha_2}{\pi \cdot (1 - \beta_1) + (1 - \pi) \cdot \alpha_1}. \quad (1)$$

*Figure 2* illustrates this equation by showing how *RR* depends on $\pi$, $\alpha_1$, and $\beta_1$ when the nominal alpha level and the statistical power of the replication studies are $\alpha_2 = 0.05$ and $1 - \beta_2 = 0.90$. It can be seen in this figure that *RR* increases gradually with $\pi$ from a minimum of $\alpha_2 = 0.05$ to a maximum of $1 - \beta_2 = 0.90$. For $\pi = 0$, the proportion of significant results can only represent false positives, so *RR* necessarily equals $\alpha_2$. For $\pi = 1$, in contrast, *RR* merely reflects the power of the replication study. As is also illustrated in this figure, *RR* grows faster when the power $1 - \beta_1$ of the original studies is relatively large and their nominal alpha level $\alpha_1$ is relatively small. Note that *RR* must gradually increase with $\pi$ from $\alpha_2$ to $1 - \beta_2$ even if the power in the original study were 100%. It is also instructive to note that worst-case *p*-hacking would imply $\alpha_1 \rightarrow 1$ and $\beta_1 \rightarrow 0$. In this case it follows from *Equation 1* that *RR* approaches the line which runs from $\alpha_2$ at $\pi = 0$ to $1 - \beta_2$ at $\pi = 1$.

If *p*-hacking is performed in the original study, this would increase the Type 1 error rate above the nominal significance level $\alpha_1$ (usually 5%) to, for example, 10% or even higher. Thus, when a researcher examines a null effect, *p*-hacking increases the proportion of false positives. The extent of this increase depends on the

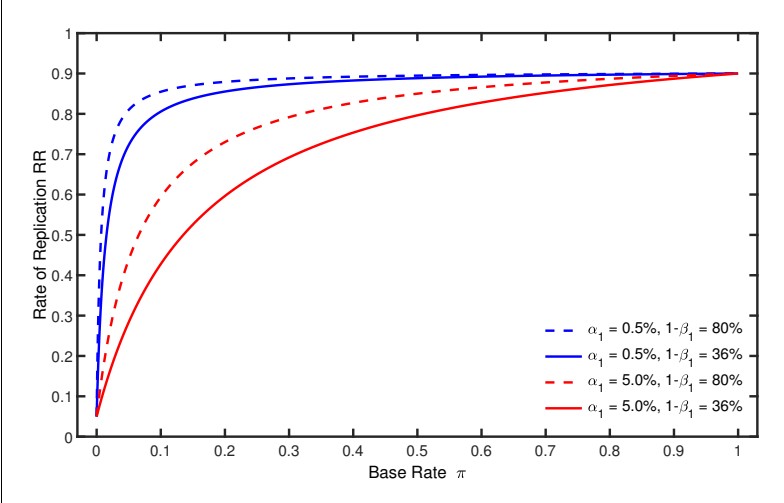

**Figure 2.** Rate of replication $RR$ as a function of base rate $\pi$. Each line represents a different combination of the nominal alpha level $\alpha_1$ and the statistical power $1 - \beta_1$ used by the original studies. The nominal alpha level and the power of the replication studies were always $\alpha_2 = 5\%$ and $1 - \beta_2 = 90\%$.

Type 1 error probability and the statistical power can be computed. These values are then inserted into *Equation 1*, which allows one to evaluate the effects of base rates and *p*-hacking on the replication rate, for both true and null effects. In addition, we examined the effects on $RR$ of different levels of $\alpha_1$ and statistical power, because—as mentioned above—several researchers have recently suggested lowering the $\alpha$ level or increasing power in order to increase the replicability of scientific results (*Benjamin et al., 2018*; *Button et al., 2013*). This allows one to judge how these suggested measures would combat low replicability and to compare their effects with those of *p*-hacking and base rate.

## Selective reporting of significant studies

It has been often suspected that researchers tend to selectively report studies that yield positive results, that is, results that are in accordance with the researcher's hypothesis (e.g., *John et al., 2012*; *Rosenthal, 1979*; *Simmons et al., 2011*; *Zwaan et al., 2018*). As noted earlier, this tendency will increase the number of reported false positives if researchers publish only the significant outcomes. This section models this *p*-hacking strategy and examines how it would influence the replication rate.

As a specific example, suppose that a researcher runs a series of experiments, each of which uses a slight variation of the same basic paradigm. This researcher terminates the series when a significant result emerges in support of the researcher's hypothesis, and in this case the researcher tries to publish that result. However, if no significant result is obtained after conducting $k$ experiments, the researcher abandons the project and concludes that the hypothesis is false. Thus, this researcher has $k$ studies providing opportunities to test the hypothesis, and it would be misleading about the overall $\alpha$ level to publish only the significant outcome but not mention the non-significant attempts (*Francis, 2014*).

To model this scenario more concretely, assume that the researcher computes a *z*-value for the outcome of each experiment and considers the outcome to be statistically significant if any *z*-value exceeds a pre-specified criterion $c$ (e.g., the critical *z* value of 1.96). In general, the probability of rejecting $H_0$ can be computed for $k \geq 1$ with

details of the *p*-hacking strategy that is used, as we examine in detail below for different strategies. However, and crucially for the analyses that will follow, when a true effect is present, *p*-hacking also increases the nominal power $1 - \beta_1$, for example, from 0.20 to 0.40 (i.e., power inflation, as mentioned above). With respect to the overall replication rate $RR$, this increase in power tends to compensate for the increased Type 1 error probability, making it difficult to determine intuitively how *p*-hacking would affect the replication rate $RR$. Fortunately, however, *Equation 1* can be used to assess this issue quantitatively.

Besides assessing the effect of these factors on replicability, we will also report computations of the rate of false positives $FPR$, which is the proportion of false positive results among all significant results within a research area (sometimes also called *false discovery rate* or *false positive report probability*)

$$FPR = \frac{(1 - \pi) \cdot \alpha_1}{(1 - \pi) \cdot \alpha_1 + \pi \cdot (1 - \beta_1)}. \quad (2)$$

In discussions about replicability—particularly replicability of published research findings—researchers often focus on this proportion (*Button et al., 2013*; *Pashler and Harris, 2012*) under the assumption that true positives are replicable but false positives are not. Therefore, it seems useful to include this rate in the analyses.

In the following, we model each of the four common *p*-hacking strategies that were described above. For each strategy, the inflated

$$\Pr(\text{Reject } H_0) = 1 - \prod_{i=1}^{k} \Pr(Z_i \leq c), \qquad (3)$$

because the outcomes of the $k$ experiments are statistically independent if a new sample is recruited each time.

*Figure 3* and *Figure 3—figure supplement 1* depict the probability of rejecting $H_0$ for two- and one-sample tests, respectively, as a function of $k \in (1, 2, 4, 6, 8)$, $\alpha \in (5\%, 0.5\%)$, and effect size $d \in (0.0, 0.2, 0.5, 0.8)$. (Appendix 1 contains a detailed description of both tests.) In these examples, the group size is assumed to be $n = 20$ (i.e., total $n = 40$ for a two-sample test), a value that is typical for psychological research (*Marszalek et al., 2011*, Table 3), though there is evidence that sample sizes have increased recently in the field of social-personality psychology (*Fraley and Vazire, 2014*; *Sassenberg and Ditrich, 2019*). The lines for $d = 0$ depict the effective Type 1 error probability. Of course, this probability is equal to $\alpha$ for $k = 1$, but it increases with $k$ because of the greater number

of opportunities for getting a significant result by chance when more studies are conducted. This increased Type 1 error probability is problematic because it tends to decrease replication rates (*Benjamin et al., 2018*). In the worst of these cases, the inflated Type 1 error rate attains a value of about 0.34 with $\alpha = 5\%$ and $k = 8$. As one expects, decreasing the nominal $\alpha$ level from 5% to 0.5% substantially diminishes the Type 1 error probability and thus correspondingly diminishes the probability of obtaining a false positive (*Benjamin et al., 2018*). Even for $k = 8$ the Type 1 error rate would only be about 0.04 with this smaller nominal $\alpha$ level. It must be stressed, however, that a larger sample would be required for $\alpha = 0.5\%$ than for $\alpha = 5\%$ to achieve the same level of statistical power in both cases (*Benjamin et al., 2018*).

The lines for $d > 0$ reveal the statistical power to reject $H_0$ when it is false. When researchers follow good scientific practice, the statistical power associated with each value of $d$ can be seen at $k = 1$. As is well known, power generally increases with $d$, and it is larger with $\alpha = 5\%$

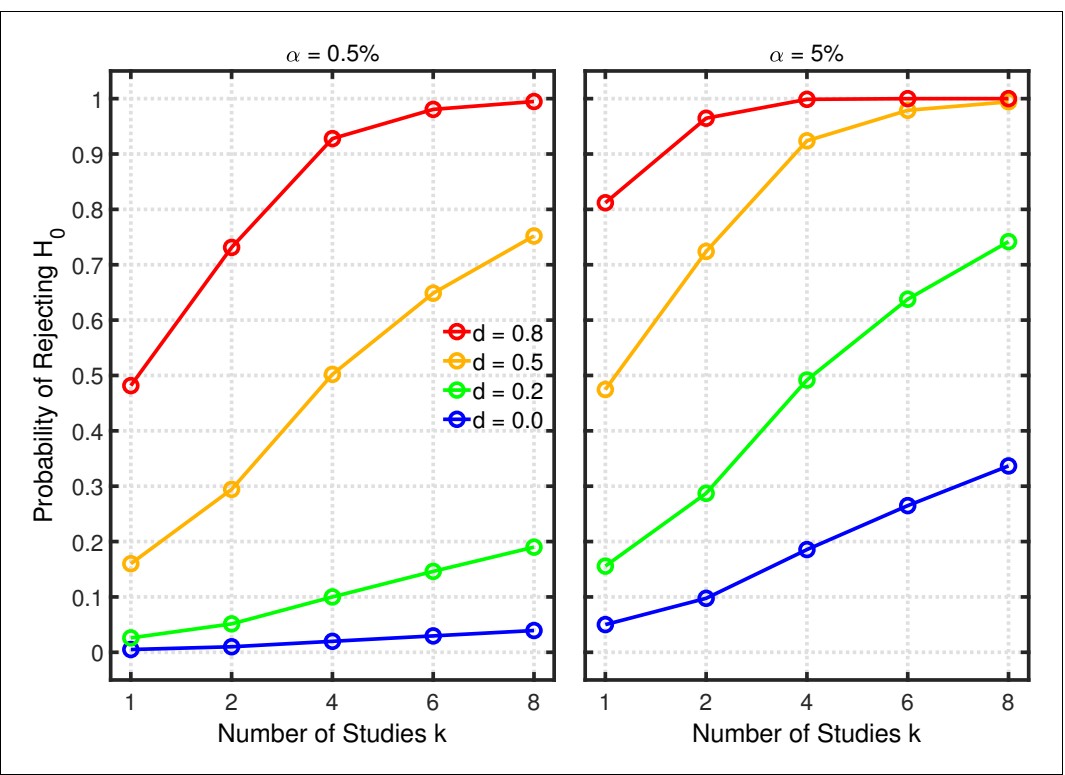

**Figure 3.** Selective reporting of significant studies. Each panel depicts the probability of rejecting $H_0$ in at least one study as a function of the number of studies $k$, nominal $\alpha$ level, and effect size $d$ for a two-sample test with $n = 20$ participants in each sample.

The online version of this article includes the following figure supplement(s) for figure 3:

**Figure supplement 1.** Selective reporting of significant studies.

than with $\alpha = 0.5\%$. For the present purposes, however, the most important aspect of the $d>0$ lines is the strong increase of statistical power with $k$, which can be seen in both panels, especially when the single-experiment power is well below one. Since replication rates increase with power (**Button et al., 2013**; **Button and Munafò, 2017**), this power inflation will tend to compensate for the increased Type 1 error rate with respect to the overall influence of selective reporting on replication rate. It is therefore necessary to use a quantitative model to assess the net effect of this practice on the replication rate.

Using the above probabilities of rejecting $H_0$, the proportion of false positives associated with this $p$-hacking scenario can be computed from *Equation 2*. *Figure 4* and *Figure 4—figure supplement 1* highlight the false positive rate as a function of $d$, $\alpha$, and $k$. Dashed lines show the rates for researchers engaged in $p$-hacking. For comparison, the solid lines depict the rates for researchers who follow good scientific practice by just running a single experiment and reaching a conclusion based on its outcome (i.e., $k = 1$). The rates for these researchers were also computed with *Equation 2* by inserting the nominal value of $\alpha$ for $\alpha_1$ and the single-experiment power for $1 - \beta_1$.

Several effects can be observed in *Figure 4* and *Figure 4—figure supplement 1*: (a) As one expects, the false positive rate decreases from one to zero with increasing $\pi$, because the proportion of true effects among all significant effects becomes larger when $\pi$ increases (e.g., *Ioannidis, 2005b*; *Wilson and Wixted, 2018*). (b) Not surprisingly, the false positive rate becomes smaller when power increases due to larger $d$ (*Ioannidis, 2005b*). (c) Most interestingly and surprisingly, the increase in false positives produced by $p$-hacking is more pronounced with larger $d$, where statistical power is higher. This is presumably because $p$-hacking cannot increase statistical power much when it is already high (i.e., when $d$ is large), so there is little power inflation to compensate for the increased Type 1 error rate. Nevertheless, the effect of $p$-hacking is far from dramatic for $k = 2$, although it can be quite prominent for larger values of $k$, especially with small base rates.

*Figure 5* and *Figure 5—figure supplement 2* depict replication rates computed using the same parameters as in the previous figures. In addition, *Figure 5—figure supplement 1* and *Figure 5—figure supplement 3* augment these figures and specifically focus on decrease in RR

(i.e., "shrinkage") caused by $p$-hacking. Three features of these computations are especially noticeable. (a) Successful replication depends strongly on the base rate. As one might expect, all rates converge to the statistical power $1 - \beta_2 = 0.90$ of the replication study, because when all significant effects are real, the replication rate simply reflects the statistical power of the replication study, whether $p$-hacking was involved in the first study or not. (b) The effect of $p$-hacking is modest for high base rates, for the smaller $\alpha$ level, and interestingly also for smaller effect sizes and hence for low statistical power. (c) As emphasized by *Benjamin et al., 2018*, the replication rate is considerably larger for $\alpha = 0.5\%$ than for $\alpha = 5\%$, especially for small base rates.

In summary, the above analysis casts doubt on the idea that this $p$-hacking strategy is a major contributor to low replicability, even though it seems to be one of the most frequent QRPs (e.g., *John et al., 2012*). Instead, it seems that using this strategy would have little effect on replicability except in research scenarios where true effects were rare but there was high power to detect them when they were present. The strongest trends suggest that a low base rate of true effects is the major cause of low replicability (*Wilson and Wixted, 2018*), since changes in base rate can cause replication rates to range across nearly the full 0–1 range.

## Failing to report all dependent measures

Failing to report all of a study's dependent measures seems to be another common QRP (*Fiedler and Schwarz, 2016*; *John et al., 2012*). In this section, we analyze how this practice would affect the rate of replicating statistically significant results. In order to model this scenario, we assume that a researcher conducts a study to test a certain hypothesis using control and experimental conditions. After data collection, however, the researcher only reports the outcomes of those dependent measures whose tests surpass the statistical significance threshold and thereby confirm the proposed hypothesis. As examples, multiple dependent measures are usually measured and statistically evaluated in neurosciences and medical research, raising concerns about Type 1 error rates in those fields (e.g., *Hutton and Williamson, 2002*; *Vul et al., 2009*).

We again employed $z$-tests to model this scenario. Let $Z_1, \ldots, Z_k$ be the outcomes for all $k$

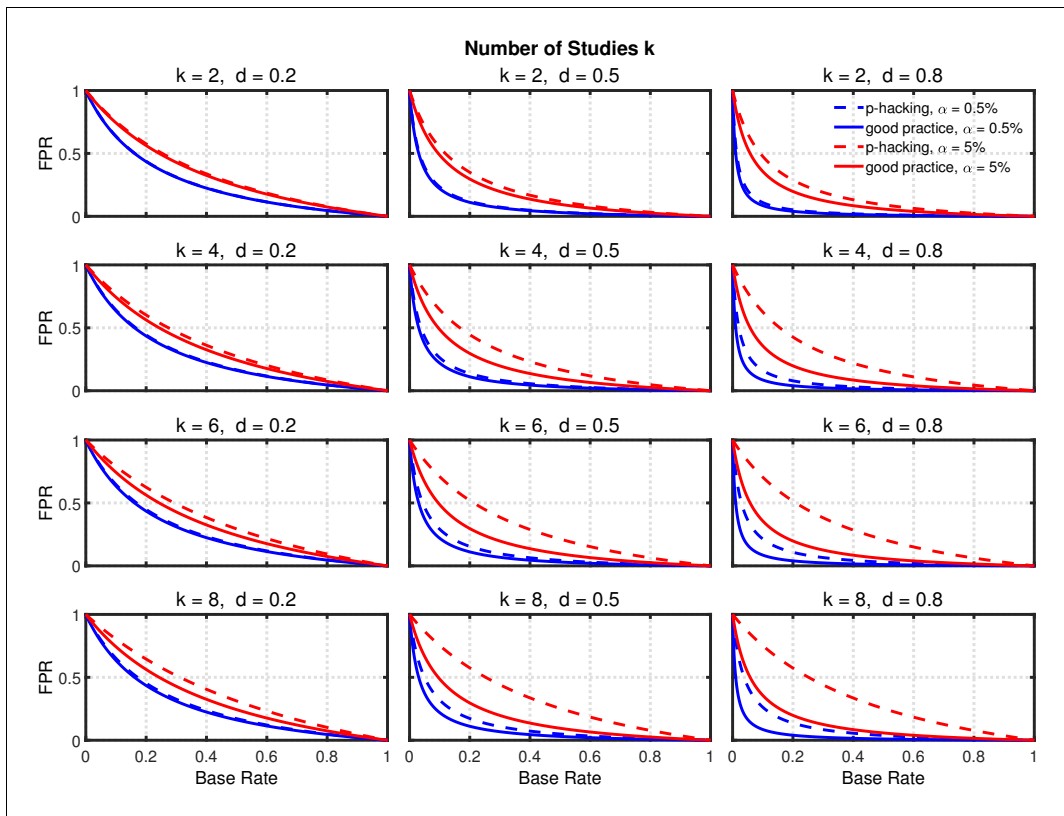

**Figure 4.** Selective reporting of significant studies. False positive rate (FPR) as a function of base rate $\pi$, number of studies $k$, effect size $d$, and nominal $\alpha$ level (0.5% or 5%). The nominal $\alpha$ level and power of the replication study are $\alpha_2 = 5\%$ and $1 - \beta_2 = 90\%$. All results are based on $n = 20$ per group. Dashed lines give the results for $p$-hacking whereas solid lines depict the results of researchers who act in accord with good scientific practice. Note that the solid lines are the same in all rows of a single column because these constant reference lines do not depend on $k$.

The online version of this article includes the following figure supplement(s) for figure 4:

**Figure supplement 1.** Selective reporting of significant studies.

dependent measures of a single study, with each $Z$-value representing the result of the control/experimental comparison for a single measure. Therefore, the probability of obtaining at least one significant result is equal to

$$\mathrm{Pr}(\mathrm{Reject}\ H_0) = 1 - P(Z_1 \leq c, \ldots, Z_k \leq c)$$

with $c$ being the critical cutoff value (see Appendix 1 for computational details). Because such measures are usually correlated across participants, our model incorporates correlations among the $Z_i$ values.

*Figure 6* illustrates the effects on Type 1 error probability (i.e., lines with $d = 0$) and statistical power (i.e., lines with $d > 0$) associated with this type of $p$-hacking. For this illustration, the pair-wise correlations of the different dependent measures were set to 0.2 and the sample size (per group) was set to 20, which are seemingly

typical values in psychological research (*Bosco et al., 2015*; *Marszalek et al., 2011*). As expected, both the Type 1 error rate and power increase with the number of dependent measures, approximately as was found with selective reporting.

*Figures 7* and *8*, and *Figure 8—figure supplement 1* show the rate of false positives, rate of replications, and the shrinkage of the replication rate, respectively, resulting from this type of $p$-hacking. These results are quite similar to those seen with the selective reporting scenario (see *Figures 4* and *5*, and *Figure 5—figure supplement 1*). In particular, both false positive rates and replication rates show strong expected effects of base rate and $\alpha$ level, as well as a clear influence of effect size, $d$. The effects of $p$-hacking are again rather modest, however, especially when the effect size is small (i.e., $d = 0.2$) so that increased power is especially helpful.

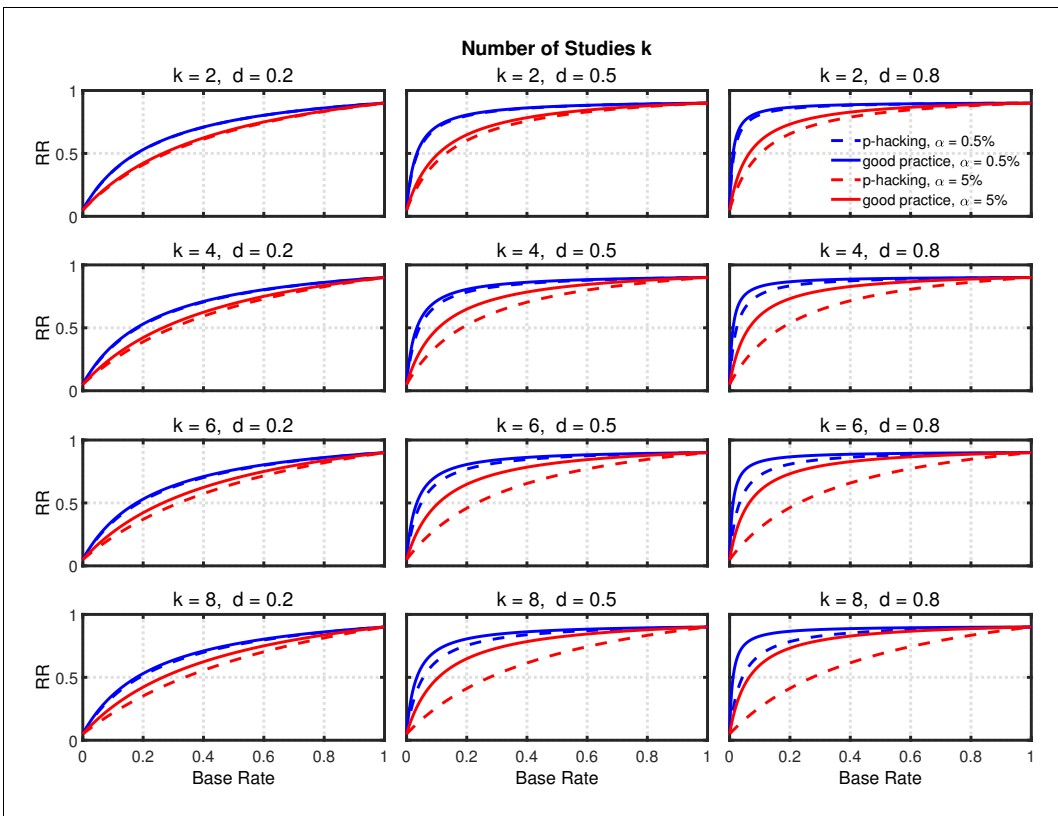

**Figure 5.** Selective reporting of significant studies. Replication rate (RR) as a function of base rate $\pi$, number of studies $k$, effect size $d$, and nominal $\alpha$ level (0.5% or 5%). The nominal $\alpha$ level and power of the replication study are $\alpha_2 = 5\%$ and $1 - \beta_2 = 90\%$. All results are based on $n = 20$ per group. Dashed lines give the results for $p$-hacking whereas solid lines depict the results of researchers who act in accord with good scientific practice. Note that the solid lines are the same in all rows of a single column because these constant reference lines do not depend on $k$.

The online version of this article includes the following figure supplement(s) for figure 5:

**Figure supplement 1.** Selective reporting of significant studies.
**Figure supplement 2.** Selective reporting of significant studies.
**Figure supplement 3.** Selective reporting of significant studies.

It should be noted that the extent of both Type 1 error rate inflation and power enhancement depend on the correlations among the different dependent measures. A correlation of zero would yield results identical to those of the scenario with selective reporting in the previous section, because in this case the outcomes for multiple dependent measures are independent just like the outcomes of multiple independent studies. In contrast, larger correlations (e.g., larger than the 0.2 used in *Figures 7* and *8* and *Figure 8—figure supplement 1*) weaken the effects of this *p*-hacking strategy, because the measures become increasingly redundant as the intercorrelations increase, and this lowers the possibility of capitalizing on chance. In other words, increasing the intercorrelations would decrease the inflation of both Type 1 error rate

and power. Moreover, increased intercorrelations would decrease the false positive rate and increase the replication rate, that is, moving the dashed lines in *Figures 7* and *8* toward the solid reference lines (see *Figure 6—figure supplement 1*, *Figure 7—figure supplement 1*, *Figure 8—figure supplement 2*, and *Figure 8—figure supplement 3* for a parallel analysis with intercorrelations of 0.8).

## Data peeking
Another frequently-used QRP is data peeking (*Fiedler and Schwarz, 2016*; *John et al., 2012*). This practice occurs when a researcher collects additional data after finding that the results of initially collected data have not yielded statistical significance. A researcher may even peek at the

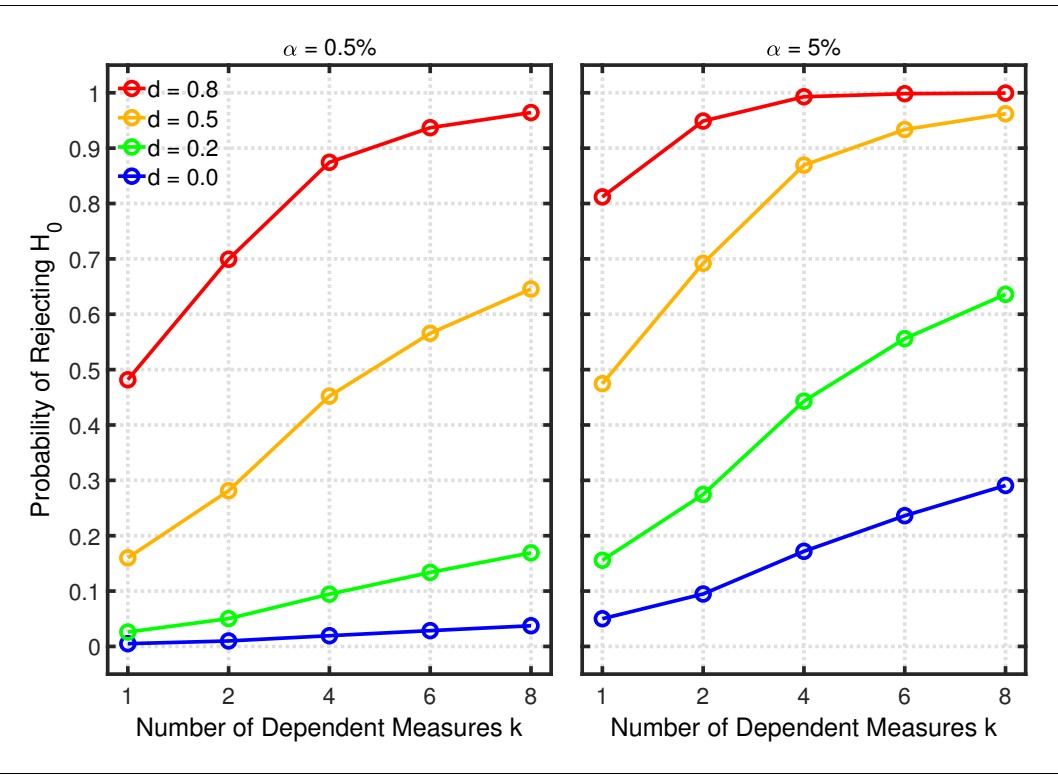

**Figure 6.** Failing to report all dependent measures. Each panel depicts the probability of rejecting $H_0$ as a function of the number of dependent measures $k$, nominal $\alpha$ level, and effect size $d$ for a two-sample test with $n = 20$ participants per group and dependent measure intercorrelations of 0.2.

The online version of this article includes the following figure supplement(s) for figure 6:

**Figure supplement 1.** Failing to report all dependent measures.

results several times and increase the sample with additional observations each time a nonsignificant result is obtained. Data collection is finally terminated only if the study yields no significant result after $k$ peeks. It is known that this practice increases the Type 1 error rate (**Armitage et al., 1969**; **Francis, 2012a**; **McCarroll et al., 1992**; **Simmons et al., 2011**; **Strube, 2006**). For example, Monte-Carlo simulations by **Simmons et al., 2011** revealed that this strategy can increase the error rate up to 14.3% with a first peek at $n = 10$ and four subsequent peeks (each time increasing the sample by 10 observations). However, this practice increases not only the Type 1 error rate but also the effective statistical power to reject a false $H_0$ (**Strube, 2006**), so a quantitative analysis is needed to determine its effect on replication rate.

An analysis similar to that of the preceding sections was conducted to examine how data peeking affects Type 1 error rates, power levels, false positive rates, and replication rates. Appendix 1 contains the computational details

of this analysis, which follows an extension of Armitage's procedure (**Armitage et al., 1969**). In brief, the probability of rejecting $H_0$ with a maximum of $k$ peeks at successive sample sizes $n_1 < n_2 < \cdots < n_k$ is again given by the multivariate normal distribution for z-tests

$$\mathrm{Pr}(\mathrm{Reject}\ H_0) = 1 - P(Z_1 \leq c, \ldots, Z_k \leq c).$$

The correlations among the different $Z_i$ values are determined by the amount of shared data used in computing them (e.g., all observations used in computing $Z_1$ are also included in the computation of $Z_2$).

*Figure 9* depicts the probability of rejecting $H_0$ for various effect sizes and two-sample tests. The abscissa represents the maximal number of peeks $k$ at which a researcher would give up recruiting additional participants. For this example, it is assumed that data peeking occurs after 10, 15, 20, 25, 30, 35, 40, or 45 observations per group. Thus, a researcher with a maximum of $k = 2$ peeks will check statistical significance the first time at $n_1 = 10$ and if the first peek does not

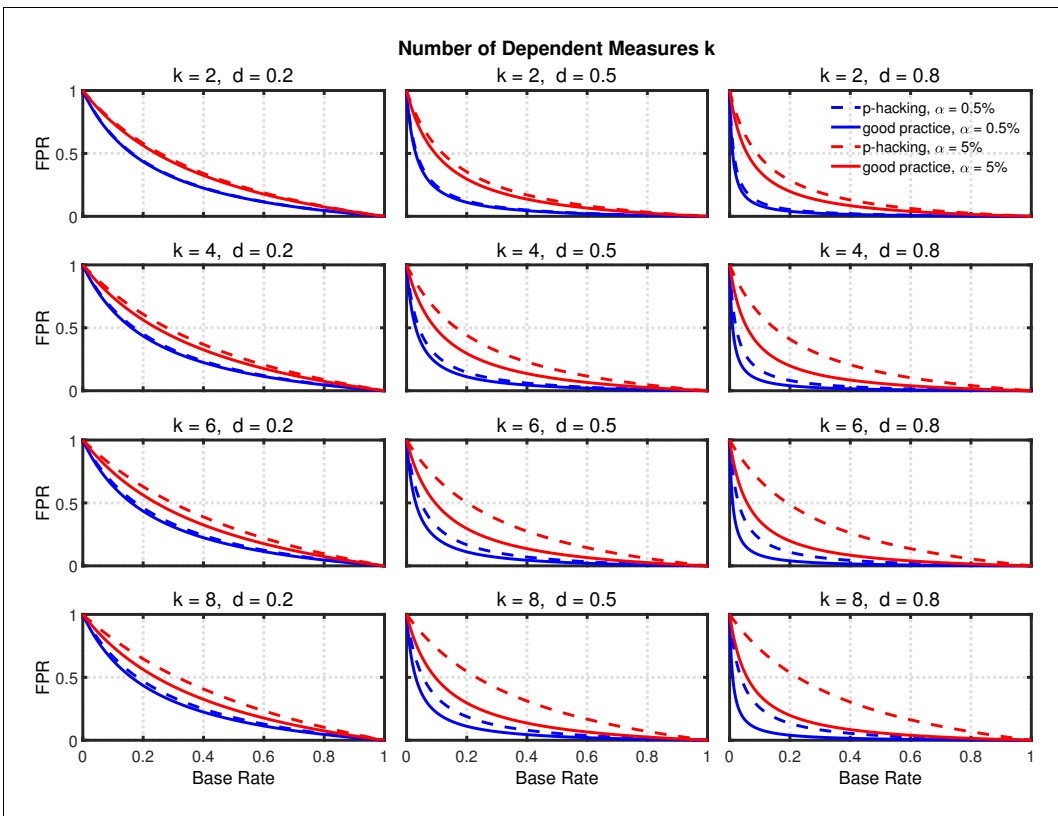

**Figure 7.** Failing to report all dependent measures. False positive rate (FPR) as a function of base rate $\pi$, number of dependent measures $k$, effect size $d$, and nominal $\alpha$ level (0.5% or 5%). The nominal $\alpha$ level and power of the replication study are $\alpha_2 = 5\%$ and $1 - \beta_2 = 90\%$. All results are based on two-sample tests with $n = 20$ per group and dependent measure intercorrelations of 0.2. Dashed lines give the results for $p$-hacking whereas solid lines depict the results of researchers who act in accord with good scientific practice. Note that the solid lines are the same in all rows of a single column because these constant reference lines do not depend on $k$.

The online version of this article includes the following figure supplement(s) for figure 7:

**Figure supplement 1.** Failing to Report all Dependent Measures.

reveal a significant result, the data will be examined a second and final time at $n_2 = 15$. For $k = 3$, data will be examined a first time at $n_1 = 10$ and—depending on the outcome of the first peek—a second time at $n_2 = 15$; if the second peek also does not reveal a significant result, a final peek occurs at $n_3 = 20$.

*Figure 9* shows quantitatively how the probability of rejecting $H_0$ increases with the maximum number of peeks. In particular, the increase can be quite strong in situations with only moderate power (e.g., $\alpha = 0.5\%$ and $d = 0.8$) due to the extra chances of detecting the true effect. In contrast to the multiple dependent measures with intercorrelations of 0.2 as discussed in the previous section, the Type 1 error rate inflation is smaller in the present case, because $Z_1, \ldots, Z_k$ are more strongly correlated under this scenario (cf. the correlation matrix in Appendix 1).

Given the probabilities of rejecting $H_0$, the replication rate and false positive rate are again computed using *Equations 1 and 2*, respectively. The results with respect to the false positive rate (*Figure 9—figure supplement 1*) and the replication rate (*Figure 10* and *Figure 10— figure supplement 1*) are quite similar to those of the preceding scenarios. We compare this $p$-hacking strategy with researchers who conform to good scientific practice and thus examine the data only once at a preplanned $n$. In order to enable a conservative comparison with $p$-hackers, we used a preplanned $n$ corresponding to the maximum number of observations a $p$-hacker would try when using the indicated number of peeks (i.e., this preplanned group size would be $n = 15$ for the comparison with $k = 2$, $n = 20$ for the comparison with $k = 3$, etc.). As can be seen, the pattern of results is quite comparable to the previous scenarios. Overall, the

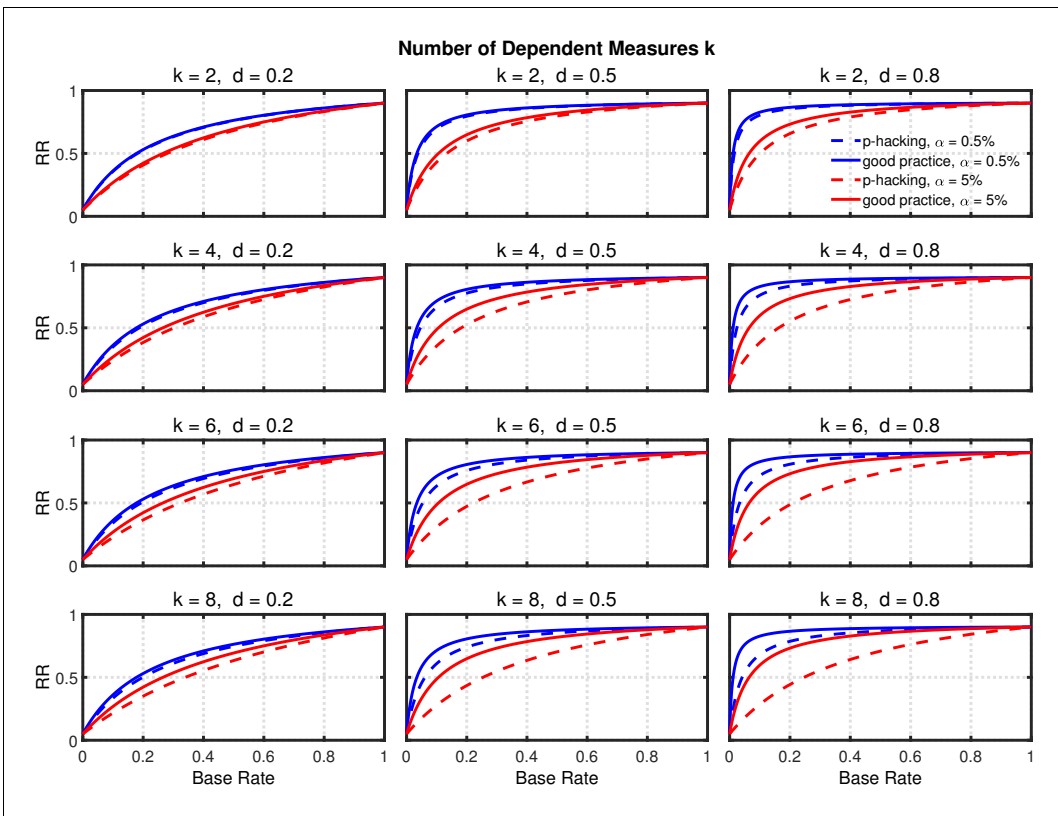

**Figure 8.** Failing to report all dependent measures. Replication rate (RR) as a function of base rate $\pi$, number of dependent measures $k$, effect size $d$, and nominal $\alpha$ level (0.5% or 5%). The nominal $\alpha$ level and power of the replication study are $\alpha_2 = 5\%$ and $1 - \beta_2 = 90\%$. All results are based on two-sample tests with $n = 20$ per group and dependent measure intercorrelations of 0.2. Dashed lines give the results for $p$-hacking whereas solid lines depict the results of researchers who act in accord with good scientific practice. Note that the solid lines are the same in all rows of a single column because these constant reference lines do not depend on $k$.

The online version of this article includes the following figure supplement(s) for figure 8:

**Figure supplement 1.** Failing to report all dependent measures.
**Figure supplement 2.** Failing to report all dependent measures.
**Figure supplement 3.** Failing to report all dependent measures.

data peeking strategy again seems to have little effect on replication rate except in research scenarios where true effects are infrequent and there is high power to detect them when they do occur, just as with selective reporting.

## Selective outlier removal

Another QRP identified by *John et al., 2012* is to analyze the same overall data set several times, each time excluding "outlier" data points identified by different criteria. The researcher may be tempted to conclude that a real effect has been found if any analysis yields a significant result, but this practice inflates the Type 1 error rate, because each of the analyses provides a further opportunity to obtain a significant result by chance. On the positive side, though, this

practice again increases power, because each of the analyses also provides a further opportunity for detecting a real effect.

Because the effects of this type of $p$-hacking are not computable, we conducted Monte-Carlo simulations to see how multiple attempts at outlier removal would affect the Type 1 error rate, power, rate of false positives, and replication rate. Specifically, we examined the common practice of excluding scores more than a given number of standard deviations from the sample mean. We simulated researchers who carried out a sequence of at most five separate analyses on a single data set. The first three analyses included only scores within 3, 2.5, and 2 standard deviations of the mean, respectively, because these limits are most commonly employed in psychological research (*Bakker and*

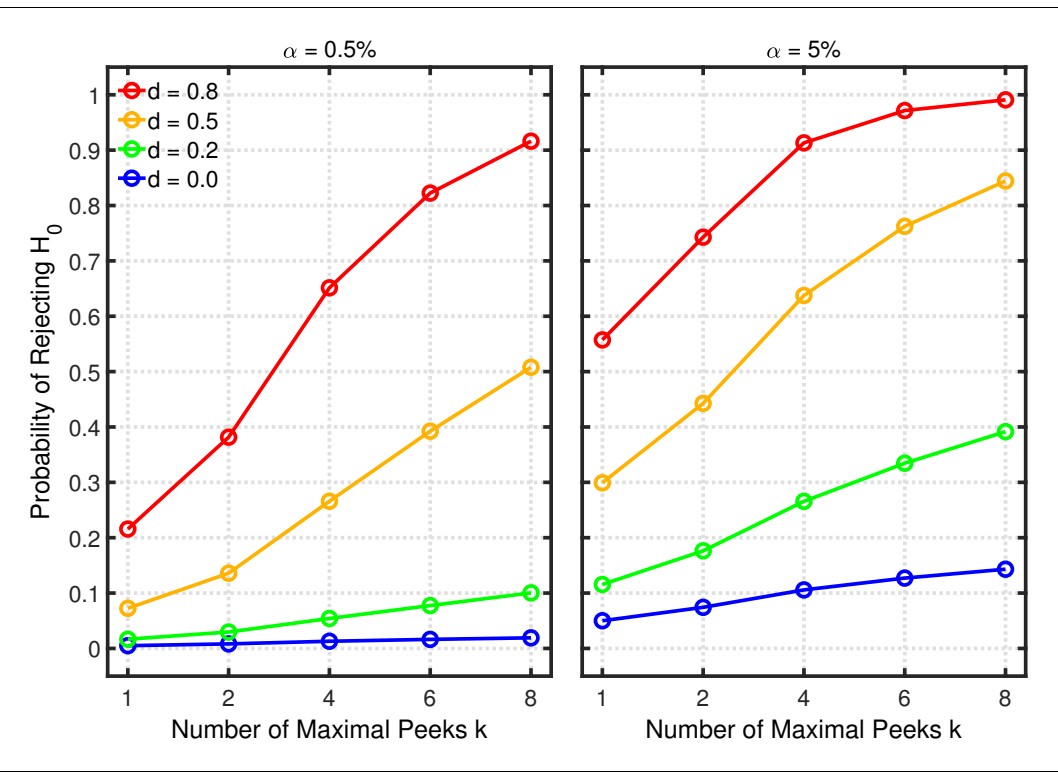

**Figure 9.** Data peeking. Each panel depicts the probability of rejecting $H_0$ as a function of the number of maximal peeks $k$, nominal $\alpha$ level, and effect size $d$ for a two-sample test.
The online version of this article includes the following figure supplement(s) for figure 9:

**Figure supplement 1.** Data peeking.

*Wicherts, 2014*). The fourth analysis used the *Tukey, 1977* "fences" method by including all scores within the range $[l = Q_1 - 1.5 \cdot (Q_3 - Q_1), u = Q_3 + 1.5 \cdot (Q_3 - Q_1)]$, where $Q_1$ and $Q_3$ are the 25 and 75% percentile points of the data set. The fifth analysis used a nonparametric test, which could potentially be used as an analysis in an attempt to minimize the influence of outliers even further.

We simulated experiments for both one- and two-sample tests, but only report the latter because the two simulations produced extremely similar results. There was a sample size of $n = 20$ per group using standard normally distributed scores and true effect sizes of $d = 0$, 0.2, 0.5, and 0.8. Researchers were modelled as using either $\alpha = 0.5\%$ or 5%, one-tailed. The nonparametric test was the Mann-Whitney $U$ test, and this test was used only if none of the previous analyses had produced significant results. We simulated 10,000 experiments with outliers by adding a random noise value to 5% of the data values, where these noise values came from a normal distribution with $\mu = 0$ and $\sigma = 10$. This simulation method has often been

adopted to model contamination effects of outliers (e.g., *Bakker and Wicherts, 2014*; *Zimmerman, 1998*).

*Figure 11* shows the probabilities of rejecting $H_0$. As with the other *p*-hacking methods, this probability increases with the number of analyses conducted, increasing the probability of a Type 1 error when $d = 0$ and increasing power when $d > 0$. *Figures 12* and *13* show the false positive and replication rates; *Figure 13—figure supplement 1* depicts the shrinkage of the replication rate. Interestingly, in some cases these measures even indicate slightly better results (i. e., lower false positive rates and higher replication rates) when researchers perform multiple analyses to remove the effects of possible outliers than when they do not. Most importantly, however, the present scenario also reveals that the major impact on the replication rate seems to come from the base rate.

The present simulations assume that researchers try to remove outliers (i.e., apply a three-sigma rule) before they perform a *t*-test. Alternatively, however, researchers might first conduct a *t*-test on all data without excluding

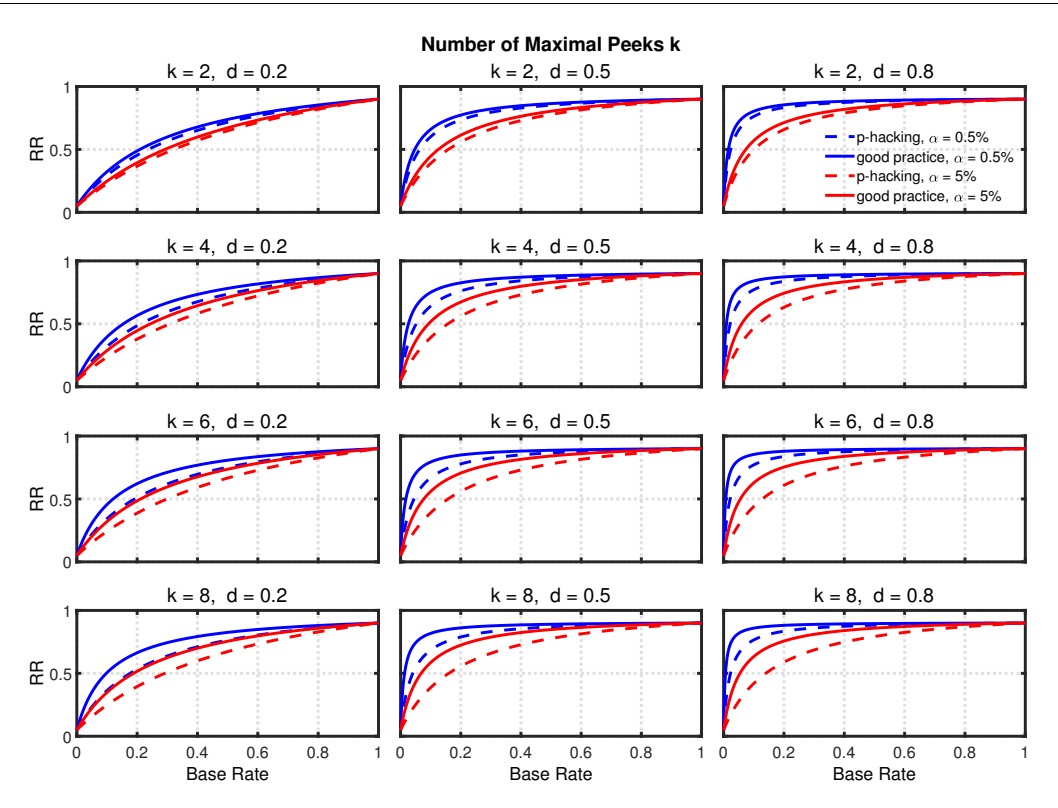

**Figure 10.** Data peeking. Replication Rate (RR) as a function of base rate $\pi$, number of maximal data peeks $k$, and nominal $\alpha$ level (0.5% or 5%). The nominal $\alpha$ level and power of the replication study are $\alpha_2 = 0.05$ and $1 - \beta_2 = 0.90$. Dashed lines give the results for $p$-hacking whereas solid lines depict the results of researchers who act in accord with good scientific practice. Note that the solid lines are the same in all rows of a single column because these constant reference lines do not depend on $k$.

The online version of this article includes the following figure supplement(s) for figure 10:

**Figure supplement 1.** Data peeking.

any extreme data points. If this test did not reveal statistical significance, they would then eliminate extreme data points before conducting one or more further *t*-tests. Under this alternative scenario, our simulations indicate that multiple analyses can produce notably better replication rates than the single analysis with all data points, apparently because the exclusion of outliers noticeably improves power relative to the analysis without exclusions. Moreover, the standard deviation of our outlier distribution was small compared to simulations of similar outlier scenarios (e.g., *Bakker and Wicherts, 2014*; *Zimmerman, 1998*). Our conclusion, of course, is that researchers should carefully examine their data for possible outliers before conducting any statistical tests, not that they should perform multiple tests with different outlier screening criteria—thereby inflating their Type 1 error rates—in order to maximize power.

Naturally, the story is different when no outliers are present in the data set. Making multiple attempts to remove outliers in this case would actually always increase the false positive rate and lower the replication rate (see *Figure 11—figure supplement 1*, *Figure 12—figure supplement 1*, *Figure 13—figure supplement 2*, and , *Figure 13—figure supplement 3* for a parallel simulation with no outliers). In fact, extreme data points in data sets without outliers appear to be especially diagnostic for testing the equality of locations between populations, as the Tukey pocket test demonstrates (*Tukey, 1959*), so throwing away extreme observations that are not outliers reduces the information in the data set.

## General discussion
The ongoing reproducibility crisis concerns virtually all sciences and naturally prompts questions

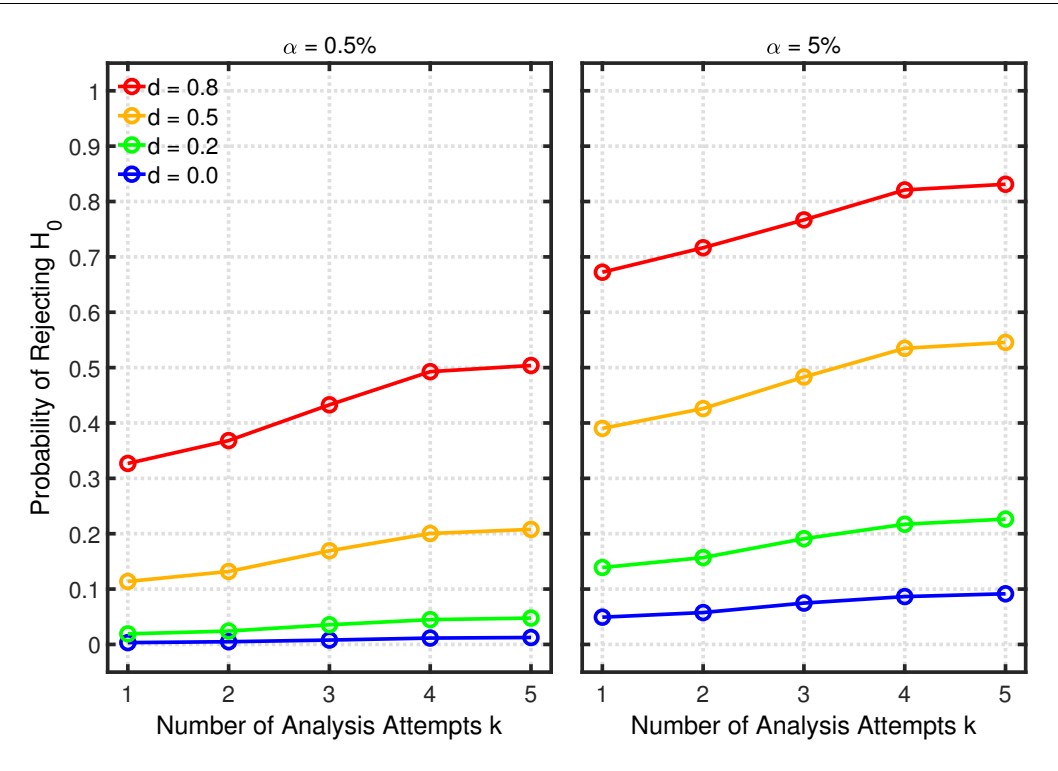

**Figure 11.** Selective outlier removal. Estimated probability of rejecting $H_0$ as a function of the number $k$ of outlier rejection methods attempted for various effect sizes $d$, and nominal $\alpha$ level (0.5% or 5%). Probability estimates were based on 10,000 simulated experiments. Simulated data included 5% outliers.
The online version of this article includes the following figure supplement(s) for figure 11:

**Figure supplement 1.** Selective outlier removal.

about how replication rates can be improved. Several measures have been advocated as ways to raise reproducibility, such as (a) preregistration of studies (*Nosek et al., 2018*), (b) increasing the transparency of research by making data and research materials publicly available (e.g., *Nosek et al., 2015*), (c) reducing $\alpha$ (*Benjamin et al., 2018*), (d) increasing statistical power (*Button and Munafò, 2017*), (e) improving statistical training (*Asendorpf et al., 2013*), (f) adopting Bayesian approaches (*Etz and Vandekerckhove, 2016*), and even (g) overhauling standard scientific methodology (*Barrett, 2020*). The variety of these proposed measures demonstrates that replication failures can result from a multitude of causes that may come into play at various steps along the "entire analysis pipeline" (*Leek and Peng, 2015*).

The present article focused on the statistical consequences of QRPs with respect to replication rate. The impacts of the various statistical factors affecting replication rate (i.e., $\alpha$, power, $\pi$, p-hacking) have typically been examined in isolation, which does not allow a complete

assessment of their mutual influence and often leads to suggestions that are difficult to implement simultaneously, such as lowering $\alpha$ and increasing power. In order to develop a better quantitative picture of the different influences on replicability, we modelled several apparently-frequent p-hacking strategies to examine their impacts on replication rate.

Our quantitative analyses suggest that p-hacking's effects on replicability are unlikely to be massive. As noted previously, p-hacking inflates the effective Type 1 error rate (e.g., *Simmons et al., 2011*), which tends to reduce replicability, but our analyses indicate that the corresponding increase in power (i.e., power inflation) substantially compensates for this inflation. Compared to the strong effect of the base rate on replicability, the reduction in replication rate caused by p-hacking appears rather small. Unsurprisingly, the impact is larger when p-hacking is more extensive (i.e., $k = 8$ rather than $k = 2$). Moreover, p-hacking affects the replication rate most when the base rate is small. This makes sense, because p-hacking is harmful

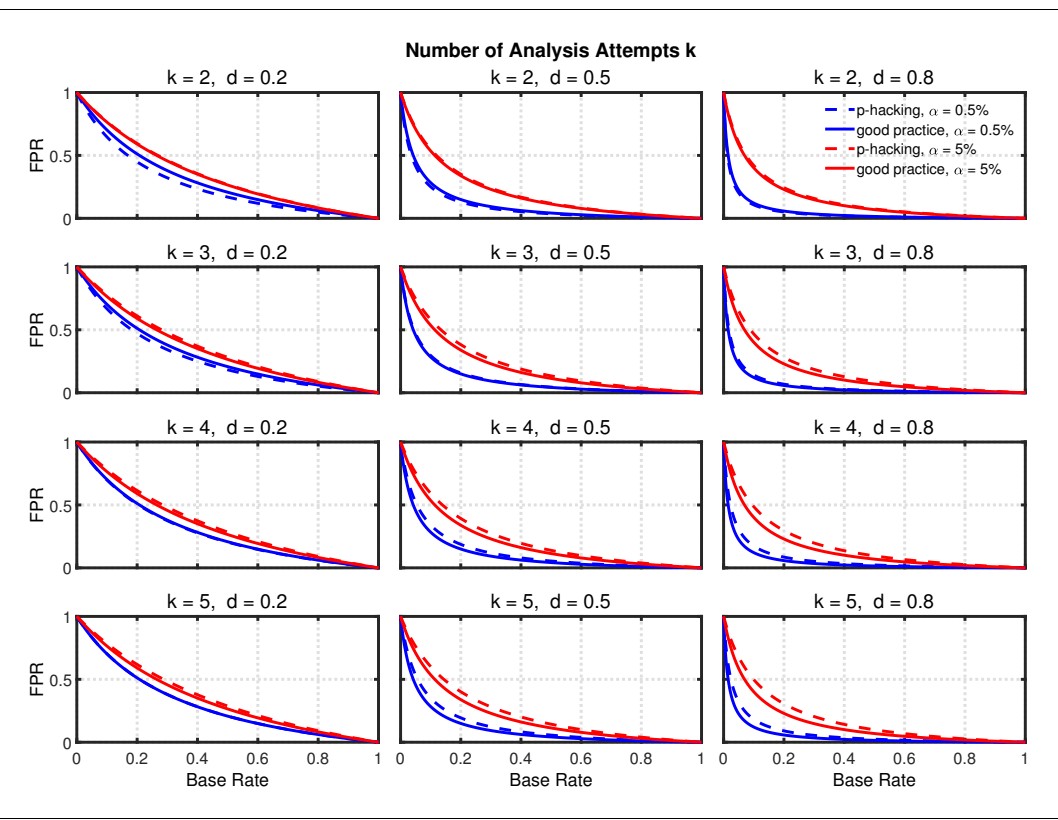

**Figure 12.** Selective outlier removal. False positive rate (FPR) as a function of the number $k$ of outlier rejection methods attempted, effect size $d$, and nominal $\alpha$ level (0.5% or 5%). The nominal $\alpha$ level and power of the replication study is $\alpha_2 = 0.05$ and $1 - \beta_2 = 0.90$. Dashed lines gives the results for $p$-hacking whereas solid lines depict the results for researchers who act according to good scientific practice. Simulated data included 5% outliers.

The online version of this article includes the following figure supplement(s) for figure 12:

**Figure supplement 1.** Selective outlier removal.

primarily when $H_0$ is true, which is more common with small base rates. The net influence of $p$-hacking on replicability appears to be smallest with small effect sizes, which is presumably the situation where $p$-hacking is most likely to be used. With small effects, the power increases associated with $p$-hacking are especially helpful for replicability. Finally and somewhat surprisingly, $p$-hacking tends to have a smaller effect on replicability when the nominal $\alpha$ level is 0.5% rather than 5%.

Of course, these conclusions are restricted to the limited extent of $p$-hacking (i.e., $k = 2, \ldots, 8$) that we examined, and more extensive $p$-hacking—or combining multiple $p$-hacking strategies—would presumably have larger effects on replicability. Nonetheless, we think that eight is a reasonable upper bound on the number of $p$-hacking attempts. The extent of $p$-hacking remains a controversial issue, with some arguing and providing evidence that ambitious $p$-hacking

is too complicated and thus not plausible (*Simonsohn et al., 2015*). Unfortunately, the exact extent of $p$-hacking is difficult to determine and might strongly depend on the field of research. For example, in areas with small effect sizes, $p$-hacking might be more extensive than in fields with medium or large effect sizes. But even without knowing the true $p$-hacking rates, our analyses are valuable because they clearly show that evidence of massive $p$-hacking is needed before one can conclude that it is a major contributor to the replication crisis. In addition, when estimating the actual effect of $p$-hacking on observed replication rates (e.g., *Open Science Collaboration, 2015*), it is important to note that the effects shown in our figures are *upper bounds* that would only be approached if nearly all researchers employed these $p$-hacking methods. If only 10% of researchers use these methods, then the overall effects on empirical replication rates would be

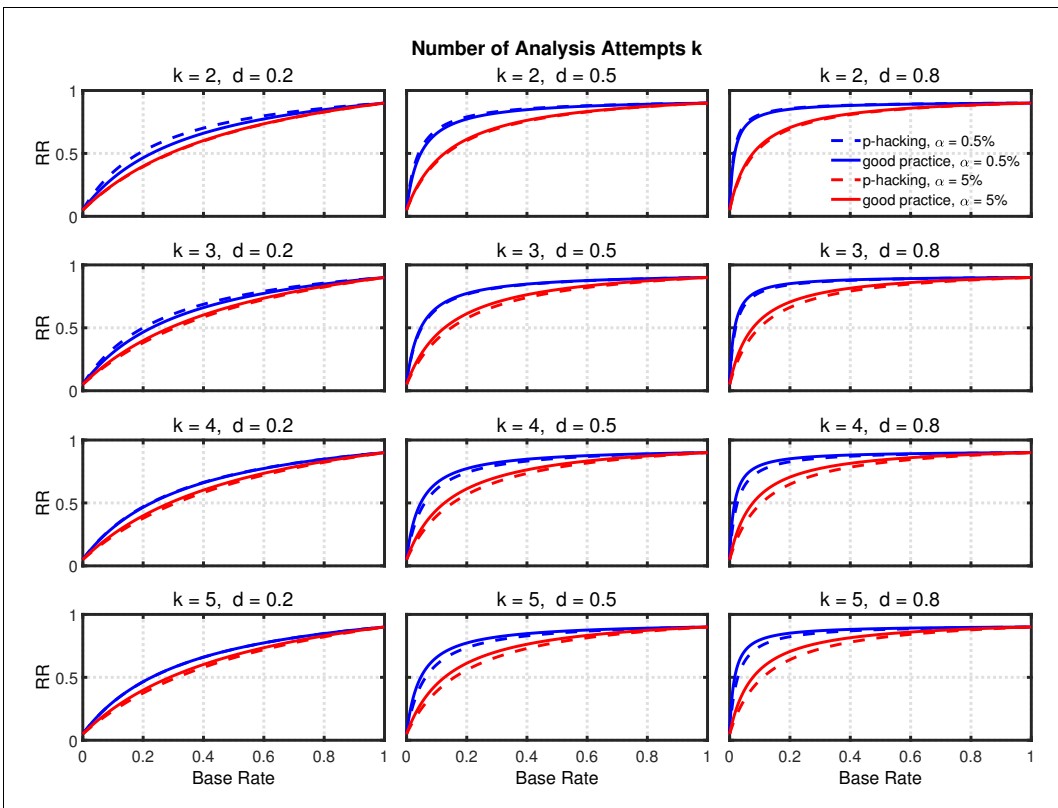

**Figure 13.** Selective outlier removal. Replication rate (RR) as a function of the number $k$ of outlier rejection methods attempted, effect size $d$, and nominal $\alpha$ level (0.5% or 5%). The nominal $\alpha$ level and power of the replication study is $\alpha_2 = 0.05$ and $1 - \beta_2 = 0.90$. Dashed lines gives the results for $p$-hacking whereas solid lines depict the results for researchers who act according to good scientific practice. Simulated data included 5% outliers.

The online version of this article includes the following figure supplement(s) for figure 13:

**Figure supplement 1.** Selective outlier removal.
**Figure supplement 2.** Selective outlier removal.
**Figure supplement 3.** Selective outlier removal.

only 10% as large as those suggested by our model. Even the highest estimates of the prevalence of QRPs are only approximately 50% (*John et al., 2012*), and these may be serious overestimates (*Fiedler and Schwarz, 2016*).

Our quantitative analysis also assumed high-powered replication studies, that is, $1 - \beta_2 = 90\%$. This replication power was chosen as the best-scenario value close to the average replication power claimed by the *Open Science Collaboration, 2015*. However, the power of the replication studies might not have been as high as they claimed. In particular, selective reporting of significant studies tends to overestimate true effect sizes, especially when these are small (*Hedges, 1984*; *Lane and Dunlap, 1978*; *Ulrich et al., 2018*), so the effect size estimates used in the power computations of the *Open Science Collaboration, 2015* may have

been too large. As a consequence, their actual power levels may have been lower than the estimated 90%. To check whether our conclusions would still be valid with lower replication power, we reran our computations using a replication power of 50%. These computations revealed that $p$-hacking would even be slightly less harmful to replication rates with 50% rather than 90% replication power.

Our analyses were based on groups size of $n = 20$ (*Marszalek et al., 2011*). Recent meta-analyses, however, have indicated an increase in sample size especially in social-personality research (*Fraley and Vazire, 2014*; *Sassenberg and Ditrich, 2019*). Therefore, one may ask whether our main conclusions still apply for larger samples. First, as discussed in the introduction, the replication rate increases gradually with base rate whether the statistical power

of the original study is low or even 100%. Therefore, even large sample studies cannot avoid low replication rates when the base rate is small. Second, because the statistical power increases with both sample size and effect size, increasing the effect size mimics what would happen if one increases the sample size. In fact, additional computations with larger samples (i.e., group size of 50) revealed no meaningful changes that would alter our conclusions.

Another limitation concerning our conclusions is that our list of p-hacking strategies was not exhaustive. For example, we did not examine the possibility that researchers might try several covariates until a significant result is obtained (e. g., *Simonsohn et al., 2014b*). As another example, suppose a researcher conducts a multi-factor analysis of variance (ANOVA) that invites the examination of multiple main effects and interactions, any one of which might be cherry picked as a "finding" in the absence of a specific a priori hypothesis. For instance, a three-factorial ANOVA allows the examination of seven potential effects (i.e., three main effects and four interactions). Assuming that all seven sources and their error terms are independent, the probability of at least one significant result when $H_0$ holds in all cases is $1 - (1 - \alpha)^7$—about 30% with $\alpha = 0.05$—which would simply emulate the multiple studies scenario that we analysed in this article. Thus, analyses similar to the present ones would be needed to analyze the consequences of these other strategies, but it would be surprising if the results were drastically different.

We supplemented the analyses reported in this manuscript by two further analyses (see Appendix 2), each of which approached the replication issue from a different angle. One supplementary analysis assessed the effect of p-hacking on power while controlling for the overall Type 1 error rate. The outcome of this analysis demonstrated that some p-hacking strategies can actually produce higher statistical power than good practice at each level of Type 1 error. This superiority can be explained by the fact that p-hacking sometimes involves the collection of additional data (e.g., as with data peeking or measuring additional variables), and in these cases the additional data can cause statistical power to increase faster than the Type 1 error rate. The other supplementary analysis compared the overall research payoff associated with good practice versus data peeking using the payoff model of *Miller and Ulrich, 2016*. This analysis showed that the expected total payoff can actually be larger with data peeking than with good practice, evidently because data peeking tends to make more efficient use of limited sample sizes when true effects are common.

If p-hacking is not a major contributor to low replicability, then what is? In keeping with previous analyses (*Dreber et al., 2015*; *Johnson et al., 2017*; *Miller, 2009*; *Miller and Ulrich, 2016*; *Wilson and Wixted, 2018*), our results suggest that low base rates of true effects—not too-large $\alpha$ levels, too-low power, or p-hacking—are most likely to be the major causes of poor replicability, so researchers concerned about replicability should pay special attention to the issue of base rates. Clearly, low base rates can lead to disappointingly low replication rates even in the absence of p-hacking (e. g., *Figures 5*, *8*, *10* and *13*, "good practice"). It follows from our analyses that research fields with inherently low base rates simply cannot improve their replication rates much by focusing exclusively on methodological issues. There are multiple lines of evidence that base rates are low in many fields (particularly those with low replication rates; e.g., *Dreber et al., 2015*; *Johnson et al., 2017*; *Miller and Ulrich, 2016*; *Miller and Ulrich, 2019*; *Wilson and Wixted, 2018*), and it will be especially challenging to increase replicability in those fields.

In principle, researchers can increase base rates by testing hypotheses that are deduced from plausible, evidence-based theories rather than by looking for effects that would be particularly surprising and newsworthy. However, practical constraints may often make it difficult to increase base rates, especially in research areas where a deeper theoretical understanding is lacking (e.g., in the search for an effective vaccine against an infectious disease). In such areas, a haphazard approach to hypothesis selection may be the only option, which naturally implies a low base rate. In combination with publication bias and p-hacking, this low base rate may make it particularly challenging to establish scientific claims as facts (*Nissen et al., 2016*).

Looking beyond replication rates, meta-scientists should consider exactly what measure of research productivity they want to optimize. For example, if the goal is to minimize false positives, they should use small $\alpha$ levels and eliminate p-hacking. If the goal is to minimize false negatives, however, they should do exactly the opposite. The major problem in statistical decision making is that one cannot maximize all of the desirable goals at the same time. Thus, focusing on only one goal—even that of

maximizing replicability—will not yield an optimal research strategy. Identifying the optimal strategy requires considering all of the goals simultaneously and integrating them into a composite measure of research productivity. One way to do this is to analyze the probabilities and payoffs for a set of possible research outcomes and to identify research parameters maximizing the expected research payoff (*Miller and Ulrich, 2016*). This analysis must also take into account how limited research resources would be used under different strategies. Other things being equal, for example, fewer resources would be needed for replication studies with $\alpha = 0.005$ than with $\alpha = 0.05$, simply because initial studies would produce fewer significant outcomes as candidates for replication.

## Conclusion

We modelled different causes (alpha level, power, base rate of true effects, QRPs) of low replication rates within a general statistical framework. Our analyses indicate that a low rate of true effects—not *p*-hacking—is mainly responsible for low replication rates—a point that is often under-appreciated in current debates about how to improve replicability. Of course, we do not wish to transmit the message that *p*-hacking is tolerable just because it might increase power when a researcher examines a true effect. As has often been discussed previously (*Simmons et al., 2011*), *p*-hacking should always be avoided because it inflates Type 1 error rates above stated levels and thus undermines scientific progress. Rather, our message is that scientists and others concerned about low replication rates should look beyond *p*-hacking for its primary causes. The current analyses suggest that even massive campaigns against *p*-hacking (e.g., researcher education, pre-registration initiatives) may produce only modest improvements in replicability. To make large changes in this important scientific measure, it will likely be necessary to address other aspects of the scientific culture. Unfortunately, that may not happen if attention and blame are focused too narrowly on *p*-hacking as a major cause of the current problems in this area.

## Acknowledgements

We thank Eric-Jan Wagenmakers and all reviewers for helpful comments.

**Rolf Ulrich** is in the Department of Psychology, University of Tübingen, Tübingen, Germany

ulrich@uni-tuebingen.de

https://orcid.org/0000-0001-8443-2705

**Jeff Miller** is in the Department of Psychology, University of Otago, Dunedin, New Zealand

miller@psy.otago.ac.nz

https://orcid.org/0000-0003-2718-3153

*Author contributions:* Rolf Ulrich, Jeff Miller, Conceptualization, Software, Formal analysis, Writing - original draft

*Competing interests:* The authors declare that no competing interests exist.

### Funding

No external funding was received for this work.

**Decision letter and Author response**
Decision letter https://doi.org/10.7554/eLife.58237.sa1
Author response https://doi.org/10.7554/eLife.58237.sa2

## Additional files
### Supplementary files
- Source code 1. Data peeking.
- Source code 2. Demo.
- Source code 3. Failing to report.
- Source code 4. Outlier rejection.
- Source code 5. Selective reporting of significant studies.
- Transparent reporting form

## Data availability

There are no empirical data because mathematical modelling was employed to assess the impact of various factors on the replication of significant results.

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

## Appendix 1

The analysis of *p*-hacking in the main article is based on one- and two-sample *z*-tests. Although researchers usually employ one- and two-sample *t*-tests rather than *z*-tests, we studied the latter tests because of their greater mathematical tractability. This should not have a big impact on the results, because *z*-tests closely resemble the results of the corresponding *t*-tests for realistic sample sizes.

### One-sample *z*-Test

This test proceeds from a random sample of $n$ observations $D_1, \ldots, D_n$ from $N(\mu, \sigma)$. Each observation $D_s$ is a single measure taken from each of the $s = 1, \ldots, n$ subjects. Let $D = \sum_{s=1}^{n} D_s$. Thus $\mathrm{E}(D) = n \cdot \mu$ and $\mathrm{SD}(D) = \sigma \cdot \sqrt{n}$. In order to test the null hypothesis $H_0 : \mu = 0$, one uses the test statistic

$$Z = \frac{D - 0}{\sigma \cdot \sqrt{n}},$$

which follows a standard normal distribution under the null hypothesis. For example, with a one-tailed test this hypothesis is rejected if $Z$ exceeds the critical value $c_\alpha$ that is associated with a pre-specified $\alpha$ level. Moreover, the effect size of this test is

$$d = \frac{\mu - 0}{\sigma}.$$

There is an alternative application of the one-sample test that is worth mentioning. In this case $D_s = X_s - Y_s$ represents a difference score for the $s$-th subject, and the dependent measures $X$ and $Y$ are most likely correlated across subjects. Consequently, the variance of $D$ is given by

$$\mathrm{Var}(D) = \mathrm{Var}(X) + \mathrm{Var}(Y) - 2 \cdot \mathrm{Cov}(X, Y).$$

If we let $\mathrm{Var}(X) = \mathrm{Var}(Y) = n \cdot \sigma^2$ and $\varrho$ be the correlation between $X$ and $Y$, the preceding expression simplifies to

$$\mathrm{Var}(D) = 2 \cdot n \cdot \sigma^2 \cdot (1 - \varrho).$$

Note that for a moderate correlation, i.e., $\varrho = 0.5$, the standard deviation of $D$ becomes $\mathrm{SD}(D) = \sqrt{n} \cdot \sigma$ and thus the test statistic under $H_0$ of this alternative is

$$Z = \frac{D - 0}{\sigma \cdot \sqrt{n}}$$

with effect size equal to $d = \mu / \sigma$. Therefore, this alternative view of the one-sample test is equivalent to the aforementioned single-variable view.

### Two-sample *z*-Test

The two-sample *z*-test proceeds from two independent samples $X = (X_1, \ldots, X_n)$ and $Y = (Y_1, \ldots, Y_n)$. To simplify matters, equal sample sizes are assumed. Without loss of generality, the first sample $X$ is a random draw from $N(\mu, \sigma)$ and the second sample $Y$ from $N(0, \sigma)$. Let $D_s = X_s - Y_s$ and $D = \sum_{s=1}^{n} D_s$; consequently, $\mathrm{E}(D) = n \cdot \mu$ and $\mathrm{SD}(D) = \sigma \cdot \sqrt{2 \cdot n}$. Thus the associated *z*-value of the statistic $D$ for testing $H_0 : \mu = 0$ is

$$Z = \frac{D - 0}{\sigma \cdot \sqrt{2 \cdot n}}.$$

In addition, the effect size of this test is

$$d = \frac{\mu - 0}{\sigma}.$$

## Multiple dependent measures

This section explains how to compute the probability of rejecting $H_0$ when a researcher assesses $k$ dependent measures for statistical significance. Assume that each of these dependent measures $D_1, \ldots, D_k$ is converted into a z-value, that is,

$$Z_i = \frac{D_i}{g \cdot \sqrt{n}} \qquad i = 1, \ldots, k \tag{1}$$

resulting in the random vector $Z = (Z_1, \ldots, Z_k)$; $g$ is equal to $\sigma$ for a one-sample test and equal to $\sigma \cdot \sqrt{2}$ for a two-sample test. This vector has a multivariate distribution $N(\boldsymbol{\mu}, \boldsymbol{\Sigma})$. The mean of each z-value is given by

$$\mathrm{E}(Z_i) = \frac{d}{\sigma_*} \cdot \sqrt{n} \qquad i = 1, \ldots, k \tag{2}$$

and the variance of each z-value must be one; $\sigma_*$ equals 1 and $\sqrt{2}$ for one- and two-sample tests, respectively.

The covariance matrix for the one-sample z-test can be derived as follows. Let the correlation coefficient of $D_i$ and $D_j$ be equal to $\varrho_{i,j} = \varrho$. Then the covariance of $Z_i$ and $Z_j$ is

$$
\begin{aligned}
\mathrm{Cov}(Z_i, Z_j) &= \mathrm{Cov}\left(\frac{D_i}{\sqrt{n} \cdot \sigma}, \frac{D_j}{\sqrt{n} \cdot \sigma}\right) \\
&= \frac{1}{n \cdot \sigma^2} \cdot \mathrm{Cov}(D_i, D_j) \\
&= \frac{1}{n \cdot \sigma^2} \cdot \varrho \cdot \sqrt{\mathrm{Var}(D_i) \cdot \mathrm{Var}(D_j)} \\
&= \frac{1}{n \cdot \sigma^2} \cdot \varrho \cdot n \cdot \sigma^2 \\
&= \varrho.
\end{aligned}
$$

Consequently, the off-diagonal elements in $\boldsymbol{\Sigma}$ are equal to $\varrho$ and those of the main diagonal are equal to 1.

For the two-sample test, the derivation of $\boldsymbol{\Sigma}$ proceeds as follows

$$
\begin{aligned}
\mathrm{Cov}(Z_i, Z_j) &= \mathrm{Cov}\left(\frac{D_i}{\sqrt{2n} \cdot \sigma}, \frac{D_j}{\sqrt{2n} \cdot \sigma}\right) \\
&= \frac{1}{2n \cdot \sigma^2} \cdot \mathrm{Cov}(D_i, D_j) \\
&= \frac{1}{2n \cdot \sigma^2} \cdot \mathrm{Cov}(X_i - Y_i, X_j - Y_j) \\
&= \frac{1}{2n \cdot \sigma^2} \cdot \left[\mathrm{Cov}(X_i, X_j) - \mathrm{Cov}(X_i, Y_j) - \mathrm{Cov}(Y_i, X_j) + \mathrm{Cov}(Y_i, Y_j)\right] \\
&= \frac{1}{2n \cdot \sigma^2} \cdot \left[\mathrm{Cov}(X_i, X_j) + \mathrm{Cov}(Y_i, Y_j)\right] \\
&= \frac{1}{2n \cdot \sigma^2} \cdot \left[\varrho \cdot \sqrt{\mathrm{Var}(X_i)\mathrm{Var}(X_j)} + \varrho \cdot \sqrt{\mathrm{Var}(Y_i)\mathrm{Var}(Y_j)}\right] \\
&= \frac{1}{2n \cdot \sigma^2} \cdot \left[\varrho \cdot n \cdot \sigma^2 + \varrho \cdot n \cdot \sigma^2\right] \\
&= \varrho.
\end{aligned}
$$

As a result, $\boldsymbol{\Sigma}$ is identical to the covariance matrix of the one-sample test.

The rejection probability

$$\mathrm{Pr}(\text{Reject } H_0 | \boldsymbol{\mu}, \boldsymbol{\Sigma}) = 1 - \mathrm{Pr}(Z_1 \leq c, \ldots, Z_k \leq c) \tag{3}$$

can be evaluated using routine *mvncdf* of MATLAB 2019a or function *pmvnorm* of the R package *mvtnorm* (*Genz, 1992*; *Genz and Bretz, 1999*; *Genz and Bretz, 2002*).

### Computing the probability of rejecting $H_0$ with multiple peeks

This section shows how to compute the probability of rejecting $H_0$ with a maximum of $k$ peeks for one- or two-sample $z$-tests. Our procedure extends the standard approach originally suggested by *Armitage et al., 1969*; (see also *Proschan et al., 2006*, p. 78), which can be used to compute the probability of rejecting $H_0$ for a one-sided test of a true null hypothesis. The extension also allows one to compute the probability of rejecting $H_0$ when the null hypothesis is false (a somewhat similar mathematical approach is provided in *Proschan et al., 2006*).

Assume that data are first checked for statistical significance (i.e., first "peek") when $n_1$ observations have been collected for a one-sample design or in each group for a two-sample design. If no statistically significant result is observed, the per-group sample size will be increased to $n_2$ and again checked for statistical significance. This strategy is repeated until a significant result is obtained or terminated after $k$ peeks when there has been no significant result at any peek. Thus, the sequence $n_1 < n_2 < \cdots < n_k$ denotes the different sample sizes at which the researcher tests the null hypothesis. In order to compute the probability of rejecting $H_0$ with multiple peeks, we let $Z_1, \ldots, Z_k$ be the $z$-values associated with the various sample sizes $n_1, \ldots, n_k$. For a one-sided test the probability of rejecting $H_0$ with a maximum of $k$ peeks is given by

$$\Pr(\text{Reject } H_0 | n_1, \ldots, n_k) = 1 - \Pr(Z_1 \leq c, \ldots, Z_k \leq c) \tag{4}$$

where $\Pr(Z_1 \leq c, \ldots, Z_k \leq c)$ is the cumulative distribution function of the random vector $Z = (Z_1, \ldots, Z_k)$ that follows a multivariate normal $N(\boldsymbol{\mu}, \boldsymbol{\Sigma})$ with $\boldsymbol{\mu} = [\mathrm{E}(Z_1), \ldots, \mathrm{E}(Z_k)]$ and covariance matrix $\boldsymbol{\Sigma}$. In addition, the cutoff $c$ corresponds to the $100 \cdot (1 - \alpha)\%$ percentile of the standard normal.

Under $H_0$, the expected means of $Z_i$ are $E(Z_i) = 0$ for $i = 1, \ldots, k$. In contrast, under $H_1$ these means are

$$\mathrm{E}(Z_i) = \frac{d}{\sigma_*} \cdot \sqrt{n_i}, \quad i = 1, \ldots, k \tag{5}$$

with $\sigma_* = 1$ for a one-sample test and $\sigma_* = \sqrt{2}$ for a two-sample test.

The covariance matrix $\boldsymbol{\Sigma}$ is completely specified by the vector $n = [n_1, \ldots, n_k]$. It can be shown that the $(i, j)$-th element for $n_j \geq n_i$ of this matrix is given by

$$\mathrm{Cov}(Z_i, Z_j) = \sqrt{\frac{n_i}{n_j}}. \tag{6}$$

In order to prove this equation, one makes use of the distributive property of covariances,

$$
\begin{aligned}
\mathrm{Cov}(Z_i, Z_j) &= \mathrm{Cov}\left(\frac{D_i}{\sqrt{n_i} \cdot \sigma_*}, \frac{D_j}{\sqrt{n_j} \cdot \sigma_*}\right) \\
&= \frac{1}{\sigma_*^2 \cdot \sqrt{n_i \cdot n_j}} \cdot \mathrm{Cov}\left(\sum_{s=1}^{n_i} X_s, \sum_{s=1}^{n_j} X_s\right) \\
&= \frac{1}{\sigma_*^2 \cdot \sqrt{n_i \cdot n_j}} \cdot \mathrm{Cov}\left(\sum_{s=1}^{n_i} X_s, \sum_{s=1}^{n_i} X_s + \sum_{s=n_i+1}^{n_j} X_s\right) \\
&= \frac{1}{\sigma_*^2 \cdot \sqrt{n_i \cdot n_j}} \cdot \left[\mathrm{Cov}\left(\sum_{s=1}^{n_i} X_s, \sum_{s=1}^{n_i} X_s\right) + \underbrace{\mathrm{Cov}\left(\sum_{s=1}^{n_i} X_s, \sum_{s=n_i+1}^{n_j} X_s\right)}_{\text{term}=0}\right] \\
&= \frac{1}{\sigma_*^2 \cdot \sqrt{n_i \cdot n_j}} \cdot \mathrm{Var}\left(\sum_{s=1}^{n_i} X_s\right) \\
&= \frac{1}{\sigma_*^2 \cdot \sqrt{n_i \cdot n_j}} \cdot n_i \cdot \sigma_*^2 \\
&= \sqrt{\frac{n_i}{n_j}}.
\end{aligned}
$$

For example, with peeks at $n = [20, 25, 30, 35, 40]$, one obtains

$$\boldsymbol{\Sigma} = \begin{bmatrix} 1.0000 & 0.8944 & 0.8165 & 0.7559 & 0.7071 \\ 0.8944 & 1.0000 & 0.9129 & 0.8452 & 0.7906 \\ 0.8165 & 0.9129 & 1.0000 & 0.9258 & 0.8660 \\ 0.7559 & 0.8452 & 0.9258 & 1.0000 & 0.9354 \\ 0.7071 & 0.7906 & 0.8660 & 0.9354 & 1.0000 \end{bmatrix}.$$

Again with $\boldsymbol{\mu}$ and $\boldsymbol{\Sigma}$, one can evaluate *Equation 4* using routine *mvncdf* of MATLAB 2019a or function *pmvnorm* of the R package *mvtnorm* (**Genz, 1992**; **Genz and Bretz, 1999**; **Genz and Bretz, 2002**).

## Appendix 2

### A comparison of research payoffs

Using the quantitative models of *p*-hacking developed in the main article, good practice and *p*-hacking can also be compared with respect to global measures of research effectiveness, in addition to the comparisons of Type 1 error rates, false positives, and replication rates. As an illustration, this section compares good practice versus the particular *p*-hacking strategy of data peeking based on the overall research payoff model of *Miller and Ulrich, 2016*.

The payoff model assumes that a researcher tests a fixed total number of participants across a large number of studies (e.g., $n_{max} = 1,000$), with each study testing either a true or a false null hypothesis (i.e., $d = 0$ or $d > 0$). Each study produces one of four possible decision outcomes: a true positive (TP) in which $H_0$ is correctly rejected, a false positive (FP) in which $H_0$ is incorrectly rejected (i.e., Type 1 error), a true negative (TN) in which $H_0$ is correctly retained, or a false negative (FN) in which $H_0$ is incorrectly retained (i.e., Type 2 error). According to the model, each outcome is associated with a given scientific payoff for the research area as a whole (i.e., $\mathcal{P}_{tp}$, $\mathcal{P}_{fp}$, $\mathcal{P}_{tn}$, $\mathcal{P}_{fn}$, in arbitrary units). The expected net payoff for any given research strategy (e.g., data-peeking, good practice) is the weighted sum of the individual outcome payoffs, with weights corresponding to the expected number of studies within that strategy multiplied by the probabilities of the different outcomes [i.e., $\Pr(TP)$, $\Pr(FP)$, $\Pr(TN)$, $\Pr(FN)$]. The numbers of studies and outcome probabilities for researchers using good practice can be computed using standard techniques (e.g., *Miller and Ulrich, 2016*), and they can be computed for data-peeking researchers using the outcome probabilities computed as described in the main article.

*Appendix 2—figure 1* illustrates expected net payoffs for a simple scenario in which positive results are either helpful or harmful to the scientific field (i.e., $\mathcal{P}_{tp} = 1$, $\mathcal{P}_{fp} = -1$), whereas negative results are basically uninformative (i.e., $\mathcal{P}_{tn} = \mathcal{P}_{fn} = 0$), and several aspects are of interest. First, as was noted by *Miller and Ulrich, 2016*, the expected net payoff increases strongly with the base rate of true effects, simply because the higher base rate increases the likelihood of obtaining true positive results. With a low base rate of true effects, the expected payoff can even be negative if the base rate is so low that FPs are more common than TPs. Second, as was emphasized by *Miller and Ulrich, 2019*, payoffs can be larger for $\alpha = 5\%$ than for $\alpha = 0.5\%$, especially when the base rate of true effects is not too small. This happens because the greater power provided by the larger $\alpha$ level outweighs the associated increase in Type 1 errors. Third, payoffs depend little on sample size, again because of a trade-off: Although larger samples provide greater power, which tends to increase payoff, they also reduce the number of studies that can be conducted with the fixed total number of participants, which tends to reduce payoff.

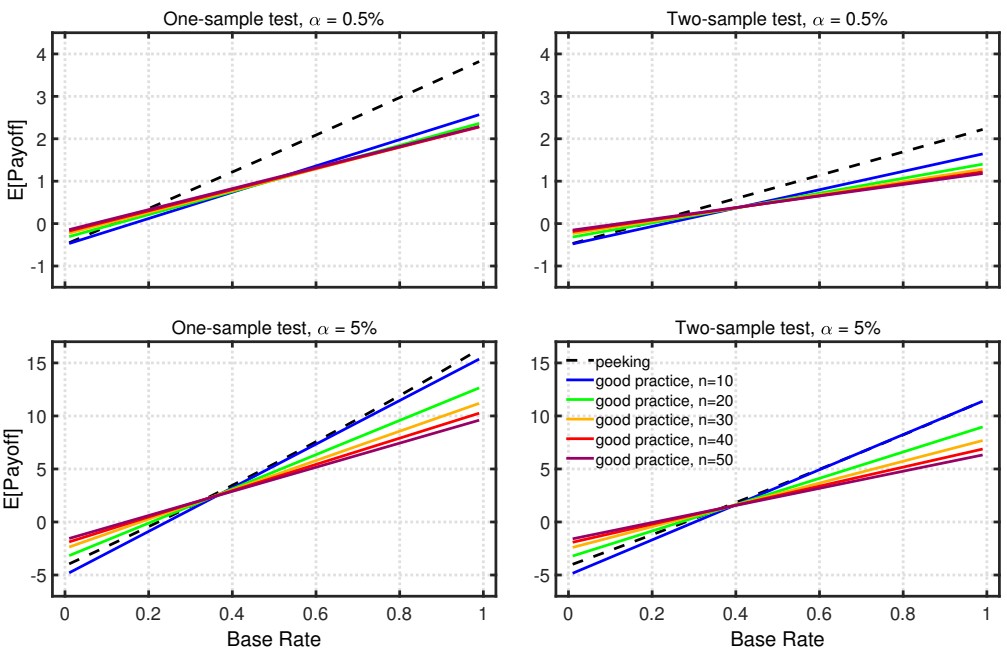

**Appendix 2—figure 1.** Expected payoff as a function of base rate and sample size. The dashed lines give the expected payoffs for researchers using data peeking at sample sizes of 10, 15, 20, 25, and 30. The solid lines give the expected payoffs for researchers who act in accord with good scientific practice and only check the data once, at the indicated sample size. The panels on the left side reflect the results for one-sample tests, whereas those on the right for two-sample tests. The upper and lower panels give the results for a nominal one-tailed $\alpha$ levels of 0.5 and 5%, respectively, with different vertical scales used because of the different ranges of payoffs for the two $\alpha$ levels. All results are based on an effect size of $d = 0.2$, individual outcome payoffs of $\mathcal{P}_{tp} = 1$, $\mathcal{P}_{fp} = -1$, $\mathcal{P}_{tn} = 0$, and $\mathcal{P}_{fn} = 0$, and a total sample size of $n_{max} = 1,000$. The results are similar for two-tailed testing (not shown).

In the present context, however, the most interesting aspect of *Appendix 2—figure 1* is that the expected payoff can be larger with *p*-hacking by data peeking than with good practice. As was noted earlier, data peeking inflates the Type 1 error rate but also increases power, and these two consequences of peeking have counteracting effects on total payoff due to the opposite weighting of TPs and FPs (i.e., $\mathcal{P}_{tp} = 1$, $\mathcal{P}_{fp} = -1$). When the base rate of true effects is large enough, the positive effects of increased power outweigh the negative effects of increased Type 1 errors. Moreover, with relatively large base rates (e.g., $\pi > 0.8$), this can be true even when the cost of an FP is much larger than the gain associated with a TP (e.g., $\mathcal{P}_{tp} = 1$, $\mathcal{P}_{fp} = -10$). Thus, under certain circumstances, data peeking would arguably be more effective than using the good-practice approach of fixing sample size in advance (e.g., *Frick, 1998*).

## Type 1 error rate versus power

Because there is an inherent trade-off between Type 1 error rate and power (i.e., larger Type 1 error rates tend to produce greater power), it is also useful to compare good practice and *p*-hacking procedures in a manner that takes both of these variables into account simultaneously. Similar comparisons are standard tools for determining the most powerful test (e.g., *Mood et al., 1974*), under the assumption that a better procedure yields higher power for a given Type 1 error rate.

*Appendix 2—figure 2* shows examples of such comparisons, plotting power versus Type 1 error rate for good practice and for each of the *p*-hacking procedures. To trace out the each line in this figure, the nominal $\alpha$ level of each procedure was varied between 0.001–0.2 in steps of 0.001. For good practice, the Type 1 error rate is simply the nominal $\alpha$ level, and power is computed using standard methods for that $\alpha$, a given effect size $d > 0$, a given sample size, and a one- or two-sample design. The analogous Type 1 error rates and power values for each of the *p*-hacking procedures

can be computed using the models described in the main article. For the *p*-hacking methods, the Type 1 error rates are greater than the nominal 0.001–0.2 $\alpha$ levels (i.e., Type 1 error rate inflation), and the curves for the different *p*-hacking methods are therefore stretched and shifted to the right. For example, with a nominal $\alpha = 0.2$—the maximum used in these calculations—the actual Type 1 error rate for multiple studies *p*-hacking is nearly 0.7.

Perhaps surprisingly, *Appendix 2—figure 2* shows that several of the *p*-hacking procedures have greater power than good practice at each actual Type 1 error rate. As an example, consider multiple studies *p*-hacking with $k = 5$ as shown in the figure. Taking inflation into account, a nominal $\alpha$ level of 0.01 produces a Type 1 error rate of approximately 0.05. For $d = 0.2$ and one-sample testing, this nominal $\alpha$ level yields power of 0.33. In contrast, using good practice with a nominal $\alpha$ level of 0.05, which of course produces the same Type 1 error rate of 0.05, the power level is only 0.23. Thus, a multiple studies researcher using the stricter nominal $\alpha$ level of 0.01 would have the same rate of Type 1 errors as the good practice researcher and yet have higher power. As a consequence of its higher power and equal Type 1 error rate, multiple studies would also produce a higher replication rate than good practice for any fixed base rate of true effects. Thus, under certain circumstances, the *p*-hacking procedures would arguably be more effective than the good-practice approach.

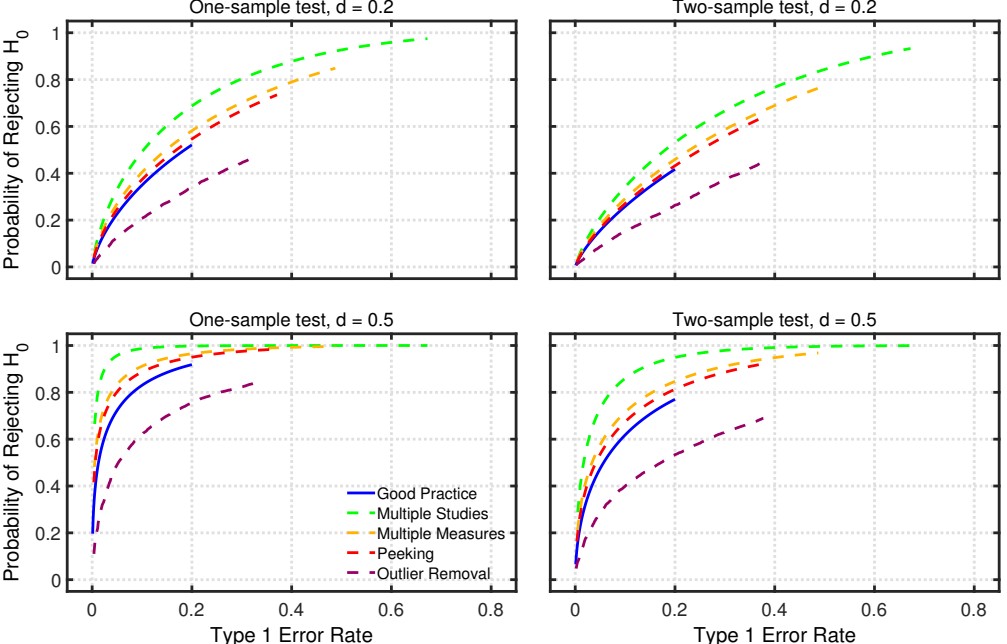

**Appendix 2—figure 2.** Power as a function of Type 1 error rate. Power for one-tailed testing as a function of Type 1 error rate for researchers using good practice or one of the four *p*-hacking procedures considered in the main article: multiple studies ($k = 5$), multiple DVs ($k = 5$ with intercorrelations of 0.2), data peeking after $n = 10$, 15, 20, 25, and 30, or multiple analyses ($k = 5$). Computations were based on a sample size of $n = 20$ (per group) for all procedures other than data-peeking.

In retrospect, it seems obvious that some types of *p*-hacking would produce higher power than good practice, because they involve collecting more data. In these examples good practice involved testing 20 participants in the example one-sample design, whereas multiple studies *p*-hacking allowed testing up to 100 participants. Data peeking also involved testing more participants—up to a maximum of 30—when that was necessary to obtain significant results. Collecting multiple DVs also provides more data because there are more scores per participant. Only multiple analysis *p*-hacking involves collecting the same amount of data as good practice, and this type of *p*-hacking yields less power than good practice at a given Type 1 error rate.

