## [Decision Letter]

Thank you for submitting your article "Meta Research: Replication of Significant Results - Modeling the Effects of p-Hacking" to *eLife*. Your article has been reviewed by three peer reviewers, and the evaluation has been overseen by the eLife Features Editor (Peter Rodgers). The following individuals involved in review of your submission have agreed to reveal their identity: William Hedley Thompson (Reviewer #2); Gregory Francis (Reviewer #3).

Summary:

The authors present an argument and simulations illustrating the impact of the replication rate of four different questionable research practices. I found the argument both interesting and convincing. The article also raises important points about interpreting replicability that are very much misunderstood by many practicing scientists. However, improvements could be made to clarify certain assumptions and aspects of the argument.

Essential revisions:

1) Equation (1) succinctly related replication rate (RR) to power and alpha. In some sense, almost everything follows from this equation. The details about how different QRPs affect alpha and beta are secondary (but useful) analyses. A further discussion of equation (1) might be worthwhile regardless of how alpha and beta take their values (through QRPs or otherwise).

2) It was not clear to me how the value for *β_2_* was selected. Based on the figure captions, I think it was just set to be *β_2_*=0.9, so 90% power. But the problem introduced by QRPs is that they tend to inflate standardized effect sizes. So, a replication of a QRP-influenced study might estimate power based on the effect size reported in the original study. Doing so will often lead to gross underestimation of power for the replication study (even though the replicator thinks they have 90% power, they might actually have 50% power). Also common, the replicator uses the same sample size as the original study, which again tends to lead to low power if the original study used QRPs. It is through these mechanisms that QRPs for an original study contribute to a low replicability rate (provided the replication study uses good practices).

3) As I read it, the headline result seemed to be that p-hacking doesn't have a large impact on replication, and therefore the explanation for surprisingly low replication must lie elsewhere. Unfortunately, the support for this claim hinges on the degree of p-hacking that one envisions, and it seems to me that the degree of p-hacking envisioned here is rather mild. My own experience suggests that most p-hacking flows from investigators' ignorance about what constitutes p-hacking, and in such cases, investigators can easily p-hack to a much greater degree than the simulations here suggest. I suspect that p-hacking investigators can easily conduct dozens if not hundreds of tests for every result ultimately published, which is a much more severe form of p-hacking than the simulations here envision. In these cases, p-hacking very well may be an important cause of non-reproducible results, thus overturning the major finding of this paper.

To be sure, the manuscript is abundantly aware that the degree to which p-hacking generates non-reproducible results depends on the degree of p-hacking, and both the results and text make that clear. So, I don't think the manuscript is 'wrong'. But, I fear that if one really wants to know how much p-hacking contributes to non-reproducible results, one has to know the extent to which p-hacked studies are indeed p-hacked.

-From the Features Editor: The article needs to explore levels of p-hacking higher than those explored in the current version and, if necessary, to revise the discussion and conclusions in the light of what these new analyses find.

Also, please consider discussing and citing the following paper:

Simonsohn et al 2015, http://dx.doi.org/10.1037/xge0000104

4) I also found the last simulation - meant to investigate the effect of removing outliers - unconvincing. As I understand it, the simulation generated hypothetical data sets that were the contaminated by outliers with a standard deviation ten-fold greater than the actual data generating process. It seems to me that this is a poor model for capturing outliers, because the outliers in this case are so anomalous that they corrupt the performance of the good-practice analysis. Thus we wind up with the head-scratching result that selective outlier removal improves FDR and RR. I don't think this set-up captures the actual hazards of selective outlier removal, and wouldn't put much stock in its results. To summarize, the simulation set-up needs to be more realistic and/or better justified.

5) A possible discussion point regarding the assumptions of the RR value. An interesting assumption in the RR value is that null/non-significant results are not replicated. And lowering the alpha threshold for statistical significance will increase the number of false negatives. So a possible outcome of focusing on improving RR with alpha thresholds that more false negatives go undetected and not replicated?

6) The authors use a group size of 20 (so total n=40) but sample size appears to be a key variable that will impact some of these measures (e.g. outlier exclusion). The researchers motivate their value by citing Marszalek, Barber, Kohlhart, & Holmes, 2011, Table 3.

First, I think the authors may be referring to table 1 to get the value of 40 (I am unable to locate the value in Table 3)?

Second, others (e.g. Fraley & Vazire 2014, 10.1371/journal.pone.0109019), found the average sample size (in social-personality) psychology research to be higher (here: 104). How dependent are the results and conclusions on the limited sample size? (especially for outlier exclusion). Also how dependent are the outlier exclusion results if more/less of the data points our outliers (currently 5% of data).

7) In Figure 3,4,6,9,11,12 and the associated text, there is no quantification about how much the "p-hacking" approach is worse and unprecise language is used, e.g. "is modest for high base rates" and the reader has to deduce the differences from the many-paneled figures. I think adding some summary numbers to the text (or an additional figure) to show the differences between methods would be useful (e.g. state RR difference when the base rate is 0.2 and 0.5 (with d=0.5, k=5) or maybe the total difference between the curves). This would be especially helpful when contrasting the differences between the "p-hacking" and "good practice" differences for the different thresholds where the reader has to deduce two differences from the graph and then compare those deduced differences in their heads.

8) One quite surprising result here was that selective outlier removal seems to increase the RR and perhaps needs a little more discussion. At the moment, a reader could read the paper and conclude that performing selective outlier removal is something that should be done to improve the RR. Is this the authors' position? If not, perhaps this should be explicitly stated.

9) The supplemental material (and a few places in the main text) suggest that QRPs might actually be favorable for scientific investigations because they increase the replicability rate. The text describes the situation properly, but I fear some readers will get the wrong impression. The favorable aspects very much depend on what a scientist wants (to avoid) out of their analyses. The supplemental material makes some claims about inflation and setting of Type I error rates and power that seem to contradict the Neyman-Pearson lemma. If not, then the multiple-studies researcher must using a larger sample size, so there are costs involved. This might be worth discussing.

10) In the Discussion the text suggests it will be difficult to increase replicability in fields with low base rates. To the contrary, I think it is easy: just increase the base rate. Scientists should do a better job picking hypotheses to test. They should not waste time testing hypotheses that would be surprising or counterintuitive. The text then goes onto discuss about how campaigns to reduce p-hacking may be ineffective. I get the point, but a field with a low base rate of hypotheses should have a low replication rate. Increasing replicability is not (or, should not) be the goal of scientific investigations.

11) The authors are familiar with some of my work on this topic (they cite several of my papers). There, the problem is not a low replication rate, but a "too high" replication rate. The problem is that if both original and replication scientists are using QRPs, then the replication rate is too high, compared to what would be expected with "good practice" analyses/experiments. In my view, this is the more serious problem with current practice, because it implies that the Type I error rate is higher than "good practice". This suggests that scientists are not doing what they intended to do. This different viewpoint struck me while reading the introduction of the paper. There it is noted that some people suggest that QRPs lead to low replication rates. But this claim never really made sense (at least not without more discussion) because QRPs increase the probability of rejecting the null; so QRPs increase the replication rate. Indeed, if the simulations were revised so that both the original and replication scientists used QRPs, there would be quite an increase in the replication rate, even when the true effect is 0.

---

## [Author Response]

[We repeat the points in the decision letter in italic, and give our responses in Roman.]

Essential revisions:1) Equation (1) succinctly related replication rate (RR) to power and alpha. In some sense, almost everything follows from this equation. The details about how different QRPs affect alpha and beta are secondary (but useful) analyses. A further discussion of equation (1) might be worthwhile regardless of how alpha and beta take their values (through QRPs or otherwise).

We agree and hence we have further discussed Equation 1 by illustrating its major features in a figure (Figure 2).

2) It was not clear to me how the value for β_2_ was selected. Based on the figure captions, I think it was just set to be β_2_=0.9, so 90% power. But the problem introduced by QRPs is that they tend to inflate standardized effect sizes. So, a replication of a QRP-influenced study might estimate power based on the effect size reported in the original study. Doing so will often lead to gross underestimation of power for the replication study (even though the replicator thinks they have 90% power, they might actually have 50% power). Also common, the replicator uses the same sample size as the original study, which again tends to lead to low power if the original study used QRPs. It is through these mechanisms that QRPs for an original study contribute to a low replicability rate (provided the replication study uses good practices).

We agree that estimated effect sizes are overestimated by replicators in such scenarios, especially when non-significant results are put into the file drawer. The power of 90% was picked as the best-scenario value close to the average replication power (i.e. 92%) claimed by the OSF project, although we acknowledge that their actual power levels may have been lower for the reasons mentioned by the reviewer. In any case, we have rerun our computations with the above suggested power of 50% and checked if our conclusions are still valid under this condition. For example, Author response image 1 shows the rate of replication for a replication power of 50% instead of 90% (as in Figure 5 of the manuscript). It can be seen in this new figure that p-hacking would even be slightly less harmful to replication rates with low-powered replication studies than with high-powered replications. Thus low-powered replication studies would not change the conclusions of our paper. We now mention this point in the Discussion.

3) As I read it, the headline result seemed to be that p-hacking doesn't have a large impact on replication, and therefore the explanation for surprisingly low replication must lie elsewhere. Unfortunately, the support for this claim hinges on the degree of p-hacking that one envisions, and it seems to me that the degree of p-hacking envisioned here is rather mild. My own experience suggests that most p-hacking flows from investigators' ignorance about what constitutes p-hacking, and in such cases, investigators can easily p-hack to a much greater degree than the simulations here suggest. I suspect that p-hacking investigators can easily conduct dozens if not hundreds of tests for every result ultimately published, which is a much more severe form of p-hacking than the simulations here envision. In these cases, p-hacking very well may be an important cause of non-reproducible results, thus overturning the major finding of this paper.To be sure, the manuscript is abundantly aware that the degree to which p-hacking generates non-reproducible results depends on the degree of p-hacking, and both the results and text make that clear. So, I don't think the manuscript is 'wrong'. But, I fear that if one really wants to know how much p-hacking contributes to non-reproducible results, one has to know the extent to which p-hacked studies are indeed p-hacked.-From the Features Editor: The article needs to explore levels of p-hacking higher than those explored in the current version and, if necessary, to revise the discussion and conclusions in the light of what these new analyses find.Also, please consider discussing and citing the following paper:Simonsohn et al 2015, http://dx.doi.org/10.1037/xge0000104

We cannot rule out the possibility that some investigators p-hack to a more considerable degree than we have assumed in our computations. The extent of p-hacking remains a controversial issue, with some arguing and providing evidence that ambitious p-hacking is too complicated and thus not plausible (Simonsohn et al., 2015, p. 1149), and even that the frequency of any p-hacking has probably been overestimated (Fiedler & Schwarz, 2016). Unfortunately, the exact extent of p-hacking is difficult to determine and might strongly depend on the field of research. For example, in areas with small effect sizes, p-hacking might be more extensive than in fields with medium or large effect sizes. But even without knowing the true p-hacking rates, our analyses are still valuable, because they clearly show that evidence of massive p-hacking is needed before one can conclude that p-hacking is a major contributor to the replication crisis. Nevertheless, for most p-hacking methods we now consider more extensive p-hacking to address this point (k to a maximum of 8). Because we cannot see how one could perform so many different selective outlier removal attempts, however, we did not consider more extensive p-hacking with this strategy. We also included this point in the General Discussion.

4) I also found the last simulation - meant to investigate the effect of removing outliers - unconvincing. As I understand it, the simulation generated hypothetical data sets that were the contaminated by outliers with a standard deviation ten-fold greater than the actual data generating process. It seems to me that this is a poor model for capturing outliers, because the outliers in this case are so anomalous that they corrupt the performance of the good-practice analysis. Thus we wind up with the head-scratching result that selective outlier removal improves FDR and RR. I don't think this set-up captures the actual hazards of selective outlier removal, and wouldn't put much stock in its results. To summarize, the simulation set-up needs to be more realistic and/or better justified.

The simulation scenario that we used to contaminate normally distributed scores has previously been employed for assessing the efficacy of outlier elimination methods (e.g., Bakker & Wicherts, 2014; Zimmerman, 1998). For example, Bakker & Wicherts and also Zimmermann sampled normally distributed scores from N(d,1), d=0, 0.2, 0.5, 0.8 with probability 0.95 and contaminated these scores with outliers that were drawn with probability 0.05 from a normal distribution with a standard deviation of 20—even larger than our standard deviation of 10 that the reviewer regards as unrealistically large. Nonetheless, in order to check how the results would change with a less extreme contamination distribution as suggested by the reviewer, we changed the standard deviation of the outlier distribution to 5. We found pretty similar results, that is, the primary factor limiting the replication rate remains the base rate (see Author response image 2 which can be compared to Figure 16 in the main text with a standard deviation of 10). Besides, the Z-score method for identifying outliers is commonly used in psychology as the meta-analysis by Bakker & Wicherts has revealed; their meta-analysis also shows that absolute thresholds values for the Z scores of 2.0, 2.5 and 3.0 are common in psychological research. Thus, our simulations capture standard methods of data analysis. In the revision, we have stressed this point.

**Author response image 2. respfig2:** 

5) A possible discussion point regarding the assumptions of the RR value. An interesting assumption in the RR value is that null/non-significant results are not replicated. And lowering the alpha threshold for statistical significance will increase the number of false negatives. So a possible outcome of focusing on improving RR with alpha thresholds that more false negatives go undetected and not replicated?

Like others modeling replication rates, we assume that researchers only try to replicate significant results. As the reviewer notes, there will be fewer of these when alpha is lowered—whether a true effect is present or *H*_0_ is true—and this effect of alpha can clearly be seen in the probability of rejecting *H*_0_ (e.g., Figure 3). Naturally, our computations of RR take this effect into account, so the effects of alpha on RR can easily be seen (e.g., Figure 5). As the reviewer rightly notes, however, reducing alpha has an additional effect that is not evident in RR: namely, fewer replication studies will be needed, because fewer positive effects will be found. We have now mentioned this fact in the General Discussion, and we thank the reviewer for the suggestion.

6) The authors use a group size of 20 (so total n=40) but sample size appears to be a key variable that will impact some of these measures (e.g. outlier exclusion). The researchers motivate their value by citing Marszalek, Barber, Kohlhart, & Holmes, 2011, Table 3.First, I think the authors may be referring to table 1 to get the value of 40 (I am unable to locate the value in Table 3)?Second, others (e.g. Fraley & Vazire 2014, 10.1371/journal.pone.0109019), found the average sample size (in social-personality) psychology research to be higher (here: 104). How dependent are the results and conclusions on the limited sample size? (especially for outlier exclusion). Also how dependent are the outlier exclusion results if more/less of the data points our outliers (currently 5% of data).

Yes, we inferred this value from Table 3 (as indicated in the previous version) but not from Table 1, as the Reviewer seems to believe. Table 3 gives the group sample size.

First, the mean of the medians in Table 3 of Marszlek et al. (2011) is 18.9. The corresponding group size in the Open Science Replication Project (considering studies with significant t-tests) is 27.5 and thus not considerably off our value of 20. It is likely that sample sizes in published articles became larger or were already larger in the field of social psychology and personality (see Sassenberg & Ditrich, 2019, Advances in Methods and Practices in Psychological Sciences). Note that sample and effect size determine the statistical power of the original study. Thus if we would employ a larger group size, this would merely increase the statistical power to an unrealistically high level --- the estimated median power level has been 36% for psychological studies and is even lower in the neurosciences. In order to keep the power of the original studies at a realistic level, we would have to reduce the effect sizes in our computations. We refrained from doing this because the effect sizes of 0.2, 0.5, and 0.8 seem appropriate theoretical choices for demonstrating the effect of p-hacking on the replication rate with small, medium, and large effects. Nevertheless, we have addressed the issue of group size in the General Discussion.

Second, Fraley and Vazire (2014) have reported an average sample size of 104 in their meta-analysis of articles published in social-personality psychology. Unfortunately, this size refers to the total sample size and not to group size. For example, consider a 2 x 2 between factorial design, then each group would be comprised of 21 subjects. Nevertheless, their article contained additional information that we found worth mentioning – for example, they estimated the average power in this area as 50% with an average effect size of d = 0.43, and that the false positive rate is (at least ) 28% for a base rate of 20%. These values fit well with our analyses. The revised paper includes Fraley and Vazire.

Third, we have rerun the simulations on outlier exclusion with group sizes of 50 (i.e., sample sizes of 100) using the new scenario (see also our response to Comment 8 below). Author response image 3 shows the result (which should be compared to Figure 16 in the main text with a group size of 20). It is clear that there are no major changes in the effects of p-hacking, alpha, or base rate, even though overall replicability has increased because of the greater power associated with larger samples.

**Author response image 3. respfig3:** 

7) In Figure 3,4,6,9,11,12 and the associated text, there is no quantification about how much the "p-hacking" approach is worse and unprecise language is used, e.g. "is modest for high base rates" and the reader has to deduce the differences from the many-paneled figures. I think adding some summary numbers to the text (or an additional figure) to show the differences between methods would be useful (e.g. state RR difference when the base rate is 0.2 and 0.5 (with d=0.5, k=5) or maybe the total difference between the curves). This would be especially helpful when contrasting the differences between the "p-hacking" and "good practice" differences for the different thresholds where the reader has to deduce two differences from the graph and then compare those deduced differences in their heads.

We like the suggestion to include an additional figure that quantifies the shrinkage of the replication rate for various level of p-hacking. In the revised paper we have added such figures along with some text that describes the resulting shrinkage.

8) One quite surprising result here was that selective outlier removal seems to increase the RR and perhaps needs a little more discussion. At the moment, a reader could read the paper and conclude that performing selective outlier removal is something that should be done to improve the RR. Is this the authors' position? If not, perhaps this should be explicitly stated.

This is certainly not our position, although our simulations indicate that selective outlier removal can improve the replication rate and lower the false positive rate. In response to this comment, we have reconsidered our outlier removal simulation. In the previous version of the manuscript, the simulated researchers always started with “no removal” and then tried more and more removal. However, “no removal” is not the best method if there are outliers. It seems perhaps a more conventional practice to remove outliers before applying a statistical test (see the meta-analysis by Bakker & Wicherts, 2014 that we cite now). Under this alternative scenario, subsequent “removals” might especially be prone to Type I error inflation and thus lower the replication rate. We have rerun our simulations under this scenario. The results indicate that multiple removals would lower the replication rate. Author response table 1 shows how we have changed the sequence of outlier removal attempts for the simulations in the revised paper. We have also modified the text in the manuscript accordingly.

Author response table 1.

We have also explicitly stated that outlier checks should be made before statistical testing and that multiple testing with different outlier criteria is unacceptable because of the increased Type 1 error rate, regardless of RR.

9) The supplemental material (and a few places in the main text) suggest that QRPs might actually be favorable for scientific investigations because they increase the replicability rate. The text describes the situation properly, but I fear some readers will get the wrong impression. The favorable aspects very much depend on what a scientist wants (to avoid) out of their analyses. The supplemental material makes some claims about inflation and setting of Type I error rates and power that seem to contradict the Neyman-Pearson lemma. If not, then the multiple-studies researcher must using a larger sample size, so there are costs involved. This might be worth discussing.

First, in the revision we have emphasized further that we do not suggest that QRPs might actually be favorable (e.g., see the Conclusions). Even in the section on outlier removal, we have now stressed that researchers should carefully examine their data before conducting any statistical tests in order to avoid inflating the Type 1 error rate. Second, the analysis in Appendix subsection “Type 1 error rate versus power” suggests that p-hacking can produce a larger power compared to good practice for the same level of Type I error. However, as the reviewer has correctly noted and as we have stated there, much larger sample sizes are the price of the potential superiority of this QRP. So the benefit actually comes from the larger samples involved with this QRP, not the QRP per se.

10) In the Discussion the text suggests it will be difficult to increase replicability in fields with low base rates. To the contrary, I think it is easy: just increase the base rate. Scientists should do a better job picking hypotheses to test. They should not waste time testing hypotheses that would be surprising or counterintuitive. The text then goes onto discuss about how campaigns to reduce p-hacking may be ineffective. I get the point, but a field with a low base rate of hypotheses should have a low replication rate. Increasing replicability is not (or, should not) be the goal of scientific investigations.

We have sympathy with this comment and agree that researchers should prefer to test hypotheses deduced from a plausible theory. However, we also see practical constraints that make it difficult to increase base rates, contrary to the reviewer’s suggestion that this would be easy. For example, consider research in clinical pharmacology aiming at discovering better medicines, such as the search for an effective vaccine against an infectious disease. Although we are not pharmacologists, we can imagine that the search for such a vaccine can be very haphazard. Pharmacological research often tests many ineffective drugs before an effective one is discovered. In such areas, the base rate could necessarily be low and only increased by a better theoretical understanding of the disease and how drugs interact with it. When such understanding is difficult to achieve, some black-box approach and the associated low base rate may be the only option. This more philosophical issue is beyond the aim of our paper, which focuses on the question of why significant results often do not replicate, but we now comment on the potential difficulty of increasing base rate in the Discussion.

11) The authors are familiar with some of my work on this topic (they cite several of my papers). There, the problem is not a low replication rate, but a "too high" replication rate. The problem is that if both original and replication scientists are using QRPs, then the replication rate is too high, compared to what would be expected with "good practice" analyses/experiments. In my view, this is the more serious problem with current practice, because it implies that the Type I error rate is higher than "good practice". This suggests that scientists are not doing what they intended to do. This different viewpoint struck me while reading the introduction of the paper. There it is noted that some people suggest that QRPs lead to low replication rates. But this claim never really made sense (at least not without more discussion) because QRPs increase the probability of rejecting the null; so QRPs increase the replication rate. Indeed, if the simulations were revised so that both the original and replication scientists used QRPs, there would be quite an increase in the replication rate, even when the true effect is 0.

We agree that an excess of positive results due to QRPs—in both original studies and replication attempts—is another potential problem that could influence the replication rate. As the reviewer notes, the replication rate will be unrealistically high if QRPs are used to induce a significant result in the replication data. This situation, however, is different from the situation in which an unbiased researcher tries to replicate the results of an original study without using QRPs, and this is the situation producing the empirically low replication rates that have alarmed many researchers. We have now made clear that our analysis focuses on this latter replication situation.